# A negative-solvatochromic fluorescent probe for visualizing intracellular distributions of fatty acid metabolites

Keiji Kajiwara[1,6], Hiroshi Osaki[1,6], Steffen Greßies[2], Keiko Kuwata[3], Ju Hyun Kim[2,4], Tobias Gensch [2,5], Yoshikatsu Sato [3], Frank Glorius [2✉], Shigehiro Yamaguchi [1,3✉] & Masayasu Taki [3✉]

Metabolic distribution of fatty acid to organelles is an essential biological process for energy homeostasis as well as for the maintenance of membrane integrity, and the metabolic pathways are strictly regulated in response to environmental stimuli. Herein, we report a fluorescent fatty acid probe, which bears an azapyrene dye that changes its absorption and emission features depending on the microenvironment polarity of the organelle into which it is transported. Owing to the environmental sensitivity of this dye, the distribution of the metabolically incorporated probe in non-polar lipid droplets, medium-polarity membranes, and the polar aqueous regions, can be visualized in different colors. Based on density scatter plots of the fluorophore, we demonstrate that the degradation of triacylglycerols in lipid droplets occurs predominantly via lipolysis rather than lipophagy in nutrition-starved hepatocytes. This tool can thus be expected to significantly advance our understanding of the lipid metabolism in living organisms.

[1] Department of Chemistry, Graduate School of Science, and Integrated Research Consortium on Chemical Sciences (IRCCS), Nagoya University, Furo, Chikusa, Nagoya 464-8602, Japan. [2] Organisch-Chemisches Institut, Westfälische Wilhelms-Universität Münster, Corrensstraße 40, 48149 Münster, Germany. [3] Institute of Transformative Bio-Molecules (WPI-ITbM), Nagoya University, Furo, Chikusa, Nagoya 464-8601, Japan. [4] Present address: Department of Chemistry, Gyeongsang National University, 52828 Jinju-daero 501, Jinju, Gyeongnam, Korea. [5] Present address: Department of Chemistry, TU Berlin, Straße des 17. Juni 135, 10623 Berlin, Germany. [6] These authors contributed equally: Keiji Kajiwara, Hiroshi Osaki. ✉email: glorius@uni-muenster.de; yamaguchi@chem.nagoya-u.ac.jp; taki@itbm.nagoya-u.ac.jp

Fatty acids (FAs) are the major components of most lipids except cholesterol. Intracellular FAs are taken up by fatty-acid transporters and readily metabolized to various bioactive lipids, such as glycerophospholipids, sphingolipids, diacylglycerols (DAGs), and triacylglycerols (TAGs), which serve as fundamental building blocks for living cells or exert various functions via binding to proteins[1,2]. These diverse fatty acid metabolites vary in their function and localization, and are strongly associated with many diseases including obesity, diabetes, and cancer[3,4]. Thus, an in-depth understanding of the transport of FAs and their metabolites in living cells can be expected to provide important biological insights.

In this context, various molecular tools that enable the direct tracing of FAs and their derivatives have been employed to understand how FAs are metabolized within cells.[5] Imaging mass spectrometry with stable isotope labeled FAs has become the gold-standard method to study the metabolism of lipids[6,7]. Alkyne-labeled FAs represent an alternative approach to analyze cellular FA metabolism[8,9]. However, their spatiotemporal resolution is insufficient for monitoring the subcellular localization of FAs and their metabolites.[5] In contrast, fluorescence imaging using fluorescent FAs that bear a small organic fluorophore on the alkyl chain is a unique tool that allows the non-invasive spatiotemporal analysis of intracellular FA dynamics[10–14]. A series of BODIPY-conjugated FAs are arguably the most widely used fluorescent lipids in biological research[11,14]. Indeed, recent studies using BODIPY 558/568 C$_{12}$, in which a BODIPY dye is appended to the ω-terminus of lauric acid, revealed how FAs are supplied to mitochondria from TAGs in lipid droplets (LDs) to produce ATP in starved mouse embryonic fibroblasts (MEFs)[11]. In contrast to the environment-insensitive BODIPY probe, FA analogues with a nitrobenzoxadiazole fluorophore are environment-sensitive, emitting weak fluorescence in aqueous media and intense fluorescence in non-polar environments[12]. Recently, a new type of fluorescent FA that emits strong fluorescence upon completion of β-oxidation has been developed[13]. Although these fluorescent FAs are promising molecular probes that provide valuable biological insights, probes for more advanced applications in the metabolic analysis of FAs demand the following characteristics: (1) A lack of interference with the metabolic pathways of FAs, including β-oxidation in mitochondria, (2) high sensitivity to the local environment polarity of the organelles in which FA metabolites are present, and (3) a neutral charge, to avoid organelle-specific localization[15]. In particular, by using solvatochromic dyes, organelle membranes and LDs have been discriminated based on the differences in the local polarity arising from lipid compositions[16]. Moreover, solvatochromic fluorescent probes allow to monitor the changes in the lipid polarity induced by starvation[17], oxidative stresses[18], and autophagy[19]. Given the recent interest in FA catabolism in autophagy upon cellular starvation, artificial FAs linked to a small and environment-sensitive fluorophore that can fluoresce with different colors in both polar and non-polar media are expected to facilitate further understanding of the complex FA metabolism.

We herein report a practical method to visualize the intracellular distribution of exogenously supplied FAs that are metabolically incorporated as lipid components, based on the environment-sensitive fluorophore 3a-azapyren-4-one (AP; Fig. 1a)[20–22]. The developed fluorescent FA probe AP-C12 (Fig. 1b) exhibits a hypsochromic shift in its absorption maximum with increasing solvent polarity, i.e., negative solvatochromism[23–25]. We discovered that AP-C12 is taken up by living cells and undergoes various metabolic events including esterification to phospholipids and TAGs as well as β-oxidation in mitochondria. Taking advantage of its characteristic photophysical properties, AP-C12 could be used for the multicolor visualization of the distribution of the FA metabolites.

Our findings using AP-C12 demonstrate that the lipolysis of TAGs in nutrition-starved hepatocytes by cytoplasmic neutral lipase is the dominant pathway for the degradation of LDs, rather than autophagic digestion.

## Results and discussion

**Evaluation of negative solvatochromic properties.** The negative solvatochromism of the AP dye originates from two different canonical structures, i.e., the neutral quinoid and aromatic zwitterion forms (Fig. 1a)[20]. The non-ionic quinoid structure should be favored in non-polar solvents, thus resulting in long-wavelength absorption and emission. In contrast, the aromatic zwitterion is favored in polar environments, resulting in a hypsochromic shift. In order to understand the nature of the negative solvatochromic behavior in depth, we first measured the photophysical properties of AP analogue **1** (hereafter AP-Me; Fig. 1c) and the annulated arylpyridine derivatives **2–6** (Fig. 1c). All the derivatives showed negative solvatochromism in both their absorption and emission, albeit that the emission maxima were less affected by the solvent polarity (Supplementary Fig. 1 and Supplementary Table 1). This indicates that the solvent interactions, and thus, the dipole moments of the dyes, in the excited state are rather small compared to those in the ground state. Even for AP-Me, which exhibits the largest negative solvatochromic shifts, the difference between the absorption maxima in toluene and in water is 78 nm, while only a 41 nm shift is observed for the emission (Fig. 1d, Supplementary Table 1). A comparison of the photophysical properties of the dyes revealed the effect of the annulated aromatic rings on the solvatochromism. Among the tested compounds, **4** bearing a thiophene ring exhibits the smallest degree of shift in its absorption wavelength (e.g., $\Delta\lambda_{abs} = 32$ nm in toluene/water). Given that the aromatic stabilization energy is a potential driving force for the adoption of the aromatic zwitterion structure in the ground state (Supplementary Fig. 2), the lower aromaticity of the thiophene ring results in less hypsochromism in polar solvents. Moreover, the hydrophobic naphthalene ring in **3** may prevent polar solvents from stabilizing the zwitterionic form in the excited state, resulting in weak negative solvatochromism in its emission. Importantly, AP-Me exhibits the highest fluorescence quantum yields ($\Phi_F$) regardless of the solvent polarity, maintaining values in the range of 0.53–0.63 (Supplementary Table 1). This implies that the K-region of the AP scaffold plays a crucial role in enhancing the fluorescence brightness. The strong fluorescence, even in water, is of great importance in visualizing the distribution of the dye throughout the cell, which is the most distinctive advantage over typical positive-solvatochromic dyes[16]. Based on these results, as well as its small molecular size and insensitivity to pH changes (Supplementary Fig. 3), we concluded that the AP dye is an excellent fluorophore for developing a fluorescent probe to analyze the FA metabolism.

**Photophysical properties in a model system.** Taking advantage of the readily modifiable 5-position of the AP ring, we prepared AP-C12 by appending saturated dodecanoic acid to make it approximately the same length as C$_{18}$-FAs such as stearic acid and oleic acid. Benzo[h]quinoline and a diazomalonate ester bearing a *tert*-butyl-protected ω-carboxylic acid were subjected to C–H alkylation–annulation using a Co(III) catalyst[21], and a subsequent deprotection by treatment with excess trifluoroacetic acid afforded AP-C12 in 51% yield (Supplementary Fig. 26 and Supplementary Note 1). Prior to cell imaging, to evaluate whether AP-C12 was capable of discriminating cell environments based on differences in lipid composition, we measured the excitation and emission spectra of AP-C12 in an aqueous buffer (1% DMSO; pH = 7.4), phospholipid vesicles (large unilamellar

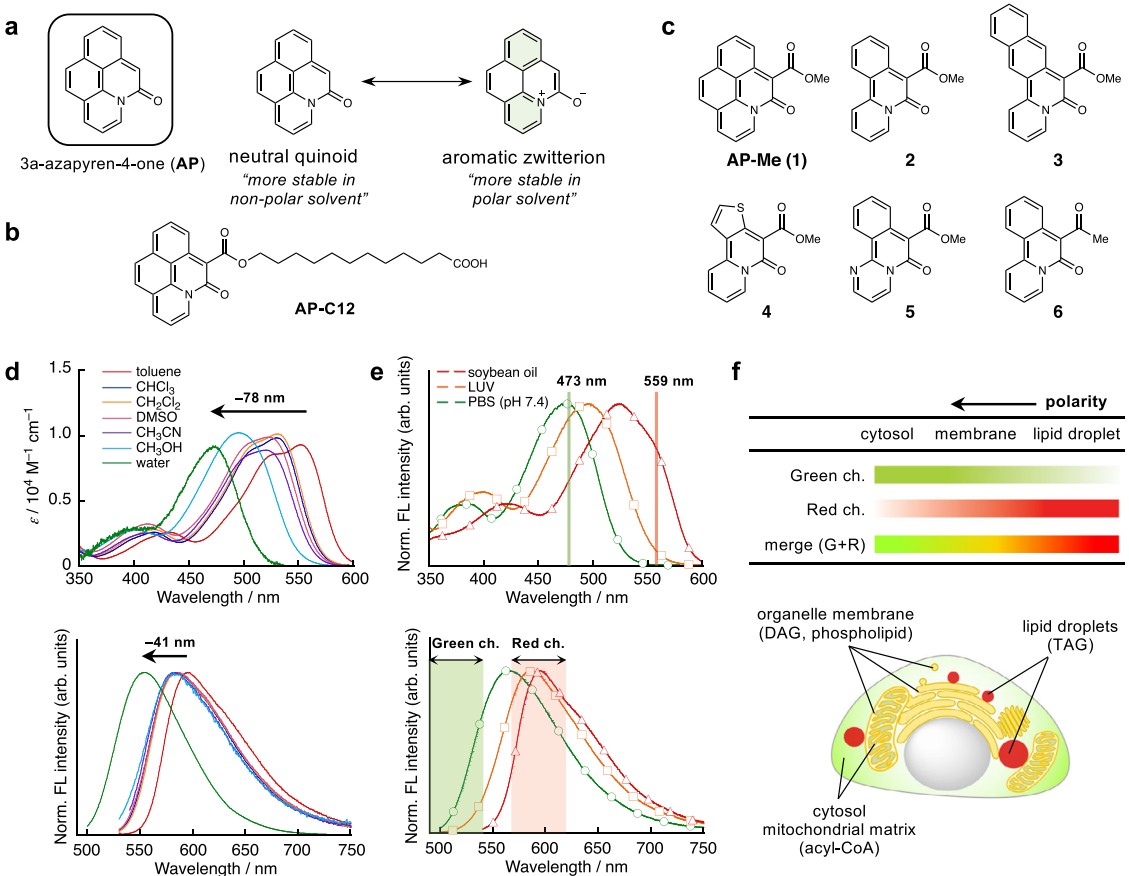

**Fig. 1 Fluorescent fatty acids based on a negatively solvatochromic dye. a** Structure of a negative-solvatochromic azapyrene (AP) dye and the equilibrium between its neutral quinoid and aromatic zwitterion forms. **b** Structure of fluorescent fatty acid probe AP-C12. **c** Structures of a simplified model compound for AP-C12, AP-Me (**1**), and related compounds annulated with different arylpyridines (**2**–**6**). **d** Absorption (top) and emission (bottom) spectra of AP-Me in various solvents. **e** Excitation (top) and emission (bottom) spectra of AP-C12 in soybean oil (1% CHCl₃), in lipid bilayers of large unilamellar vesicles (LUVs), and in PBS (pH = 7.4, 1% DMSO), which are indicated in red (triangle), orange (square), and green (circle), respectively. Green channel: $\lambda_{ex} = 473$ nm and $\lambda_{em} = 490$–540 nm; red channel: $\lambda_{ex} = 559$ nm and $\lambda_{em} = 570$–620 nm. **f** Organelles that can be detected using the green and red channels (top). Proposed pseudo-color image of AP-C12-treated cells in the merged image (bottom). Source data for this figure are provided as a Source Data file.

vesicles (LUVs) prepared from 1,2-dioleoyl-*sn*-glycero-3-phosphocholine), and soybean oil (1% CHCl₃), which are model systems that mimic the local environments of cytosol and the mitochondrial matrix, phospholipid membranes, and LDs[26], respectively. The excitation spectra and fluorescence spectra of AP-C12 showed distinct negative solvatochromism, as was observed for AP-Me in different solvents (Fig. 1e). Based on the spectral features, we decided to use the following two channels: a green channel ($\lambda_{ex} = 473$ nm; $\lambda_{em} = 490$–540 nm) and a red channel ($\lambda_{ex} = 559$ nm; $\lambda_{em} = 570$–620 nm). We expected to observe the fluorescence signals of the AP-C12 metabolites localized in cytosol, the mitochondrial matrix, and lipid membranes in the green channel, while the probes transported to lipid membranes and LDs were expected to be detected in the red channel. Consequently, merging the images recorded in these two channels (G + R) would provide a multicolor image of organelles classified according to their polarity, which would allow obtaining information regarding the subcellular localization of the AP-C12 metabolites (Fig. 1f).

**Cell staining with AP-C12.** AP-C12 showed no significant toxicity at concentrations below 10 μM, while a slight decrease in cell viability was found when the dye concentration was increased to 20 μM (Supplementary Fig. 4). In order to validate our

hypothesis that the AP-C12 metabolites exhibit different fluorescence properties in response to the environmental polarity of organelles, we first incubated adipocytes differentiated from 3T3-L1 cells (3T3-adipocytes) in the presence of AP-C12 and recorded fluorescence images in the green and red channels without washing the cells. Immediately after staining, diffuse fluorescence signals were detected throughout the cells in both channels (Fig. 2a, top). Incubation for a further 30 min resulted in an increase in the fluorescence signals from LDs in the red channel, while relatively weak fluorescence with no distinct features was observed in the green channel (Fig. 2a, middle). This suggests that AP-C12 was taken up by the cells as an exogenous FA and readily converted into the corresponding TAGs in a similar manner to natural FAs, and that the TAGs subsequently accumulated in LDs. To confirm the metabolic progression of AP-C12, 3T3-adipocytes were pretreated with 1.2 mM 2-bromooctanoate[27,28], a competitive inhibitor of diacylglycerol acyltransferase (DGAT), which catalyzes TAG biosynthesis on the ER membrane, for 30 min and then imaged after incubation with 5 μM AP-C12 for 30 min. As expected, accumulation of the AP dye in LDs was significantly decreased, while stronger fluorescence signals were observed over the entire cell in the green channel, resulting in the enhancement of green and yellow colors in the merged image (Fig. 2a, bottom). The yellow signals may be attributed to the fact

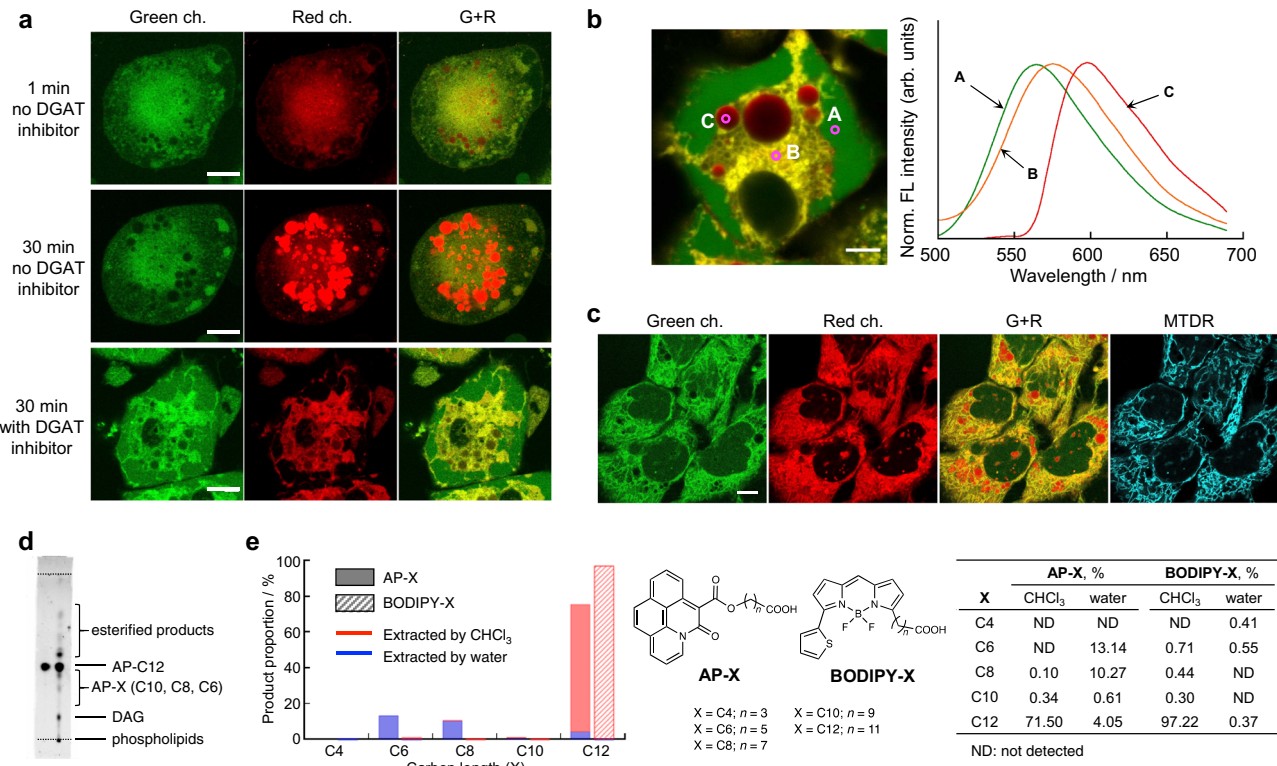

**Fig. 2 Distribution of AP-C12 metabolites in live cells. a** Confocal images of 3T3-adipocytes recorded in the green (left) and red (center) channels upon treatment with AP-C12. The merged images of the two channels are shown at the right. The images were captured 1 min (top) and 30 min (middle) after the addition of AP-C12. The images at the bottom were recorded after treatment with 2-bromooctanoate (1.2 mM) for 30 min, followed by incubation with AP-C12 for 30 min. Scale bar: 20 µm. **b** Unmixed image of 3T3-adipocytes (left) using the reference spectra of selected pixels in A, B, and C. The cells were treated with 2-bromooctanoate for 30 min prior to incubation with AP-C12. The emission spectra for the corresponding areas are shown at right. Scale bar: 5 µm. **c** Images of HepG2 cells incubated with AP-C12 and subsequently labeled with MitoTracker Deep Red$^{FM}$ (MTDR: $\lambda_{ex} = 635$ nm and $\lambda_{em} = 660$–710 nm). Scale bar: 10 µm. **d** TLC image of AP-C12 (left) and its metabolites extracted by chloroform from HepG2 cells (right). The cells were incubated with 5 µM of AP-C12 in HEPES-buffered HBSS+ for 6 h. The mobile phase is a mixture of chloroform/methanol, 9:1 (vol/vol). **e** Relative proportions (%) of the β-oxidation products (AP-X and BODIPY-X; X = C4, C6, C8, and C10) for AP-C12 (filled bar) and BODIPY 558/568-C12 (hatched bar). The products extracted with water and CHCl₃ are shown in blue and red, respectively. The results represent the average of two experiments. Source data for this figure are provided as a Source Data file.

that the DAGs derived from AP-C12 were not converted to TAGs and remained in the ER membrane.

This observation was also strongly supported by spectral imaging microscopy. The spectral image stack of the AP-C12-treated 3T3-adipocyte was obtained in the range of 468–689 nm with excitation at 458 nm and unmixed into 3 fractions, namely, cytosol, membrane, and LD, based on the emission spectra of selected pixels (Fig. 2b). The spectral features of these compartments were in good agreement with those observed in the corresponding model systems (Fig. 1e). Moreover, the resulting unmixed spectral image was comparable to the merged image shown in Fig. 2a, confirming the validity of our method using green and red channels to detect the AP signals for the analysis of FA metabolism.

We next applied AP-C12 to HepG2 hepatocytes, which are considered suitable for research of FA metabolism[29]. To produce larger LDs, the cells were preincubated in the presence of both oleic acid and palmitic acid in Dulbecco's modified Eagle's medium (DMEM) with 10% fetal bovine serum (FBS) and 4 mM glutamine (i.e., complete medium, CM). The cells were then incubated in HBSS+ (Hank's balanced salt solution with Ca²⁺ and Mg²⁺) containing AP-C12 for 1 h and imaged after labeling with MitoTracker Deep Red (MTDR). Interestingly, unlike in the case of 3T3-adipocytes, which have high TAG biosynthetic activity, network structures with intense signals were detected in

the green channel (Green ch. in Fig. 2c); these can be seen more clearly in the merged image in yellow (G + R in Fig. 2c). The co-staining result with MTDR clearly indicates that mitochondria can be visualized (MTDR in Fig. 2c), implying that AP-C12 is supplied to the hydrophilic mitochondrial matrix during nutrition starvation in HBSS[11]. It should be noted that the mitochondrial structures were observed even in the red channel, probably due to the slightly lower polarity of the local environment of AP-C12 metabolites in the mitochondrial matrix, to which the cristae membranes are highly exposed[30].

To confirm that AP-C12 is metabolically distributed in the cells, after HepG2 cells were incubated with the probe in HBSS+ for 6 h, the cell lysate was separated on a thin-layer chromatography (TLC) plate, and then imaged using a fluorescence scanner. In addition to the original spot of AP-C12 ($R_f = 0.48$), many spots corresponding to the metabolic products were newly observed (Fig. 2d). Based on the retention time of the authentic sample, AP-C6, and the reported lipid chromatography of LDs isolated from cells[31], the spots seen just below AP-C12 are ascribed to the β-oxidation products, whereas the spots appearing above AP-C12 could be attributed to TAGs and cholesterol ester containing at least one AP moiety. Moreover, the lower spot with $R_f = 0.15$ and the origin may be from DAGs and phospholipids, respectively. This result supports that the cell images observed with AP-C12 are from the metabolized probes localized in the respective organelle.

**Tracking β-oxidation products via mass spectrometry analysis**. β-oxidation of FAs in the mitochondrial matrix is an essential metabolic process for producing energy. However, fluorophores often interfere with the β-oxidation process by causing the fluorescent FAs not to be recognized as substrates or greatly slowing the reaction rate compared to that of natural FAs[11,13]. As intense signals of the AP-C12 metabolites were observed in the mitochondria, we can infer that AP-C12 is effectively consumed by β-oxidation in the mitochondrial matrix. To confirm this, we performed mass spectrometry analysis on cell lysates of HepG2 cells incubated with AP-C12 under HBSS-starved conditions.

After 1 h of incubation, the cells were harvested and subjected to an extraction using chloroform and water. The volumes of each layer were adjusted to be identical, and then fragments containing the AP moiety generated by in-source collision-induced dissociation (in-source CID) were detected using LC-MS (Fig. 2e). This method allows us to consider the peak area ratio as the abundance ratio of the AP derivatives. The mass spectrum of the organic layer showed relatively weak but distinct peaks for AP-C10 (0.3%) and AP-C8 (0.1%), which are the β-oxidation products shortened by two and four carbons, respectively, in addition to the original AP-C12 (Supplementary Fig. 5). On the other hand, in the aqueous layer, in addition to AP-C10 (0.6%), AP-C8 (10.2%) and the further β-oxidized product AP-C6 (six carbons shorter, 13.1%) were dominant, but the peak detected for AP-C4 (eight carbons shorter) was negligible (Supplementary Fig. 6). These results provide strong evidence that AP-C12 is transferred to the mitochondrial matrix, where it is readily degraded by β-oxidation. Importantly, in the culture medium, significant amounts of AP-C6 and AP-C8 were detected by MS (Supplementary Fig. 7). This may be due to the greater influence of the AP fluorophore on the nature of the FA upon shortening the carbon chain of AP-C12. That is, the AP-C12 metabolites with carbon chains shorter than six carbons were no longer recognized as a substrate for the β-oxidation cycle. Once effluxed from the cells, the short-chain AP-FAs cannot cross the membrane again via FA transport proteins (FATPs). This is also supported by the fact that no significant fluorescence signal from the AP dye was detected in the incubation medium even when the cells were cultured with the separately synthesized AP-C6 (Supplementary Fig. 8).

Next, we performed the same experiment using BODIPY 558/568 C12, which has been used in studies on FA metabolism[32]. In contrast to AP-C12, > 97% of the original BODIPY 558/568 C12 was detected in the organic layer, and the proportion of β-oxidation products was small (Supplementary Figs. 9 and 10), i.e., the β-oxidation of BODIPY 558/568 C12 did not progress as rapidly as that of AP-C12. Consequently, from the viewpoint of studying FA metabolism by β-oxidation, we concluded that AP-C12 is more suitable for evaluating the FA metabolism.

**Visualization of metabolic pathways of AP-C12.** Numerous enzymes related to lipid metabolism have been identified[33]. In general, cytotoxic free FAs taken up by cells through FATPs are immediately converted into the corresponding fatty acyl-CoAs by acyl-CoA synthetase (ACS) in the cytosol. The fatty acyl-CoAs are then transported to the ER, where TAGs are sequentially synthesized by a series of membrane-localized enzymes, including diacylglycerol acyltransferase 1 (DGAT1), which catalyzes the generation of TAGs from DAGs. The TAGs thus accumulated between the lipid bilayers of the ER eventually bud off to form LDs. The TAGs are hydrolyzed on the surface of LDs interacting with the mitochondria by adipose triglyceride lipase (ATGL) to release free FAs, if necessary. After the FAs thus regenerated are converted into acyl-CoAs, they are transported into the

mitochondrial matrix through modification with carnitine palmitoyltransferase I (CPT1) on the mitochondrial outer membrane. Herein, we imaged the distributions of the AP-C12 metabolites in cells treated with known inhibitors for the FA metabolic pathways described above (Supplementary Fig. 11).

In the following experiments, we used HepG2 cells that were treated with each inhibitor in CM for 18 h and then incubated for 6 h in HBSS+ containing 5 μM AP-C12 and 0.5 mM oleic acid[34] without changing the inhibitor conditions (Fig. 3a). First, as the control, cells without any inhibitor were imaged and analyzed. The observed image clearly shows relatively large LDs in red (Fig. 3b, control). In the corresponding scatter plot of the signal intensities (green vs. red), the pixels with both green and red components are indicated in yellow; these were attributed to the AP dye in the membrane (Fig. 3c).

When treated with triacsin C (TC), a long-chain ACS-specific inhibitor, the fluorescence intensities both in the green and red channels decreased considerably compared to those in the control. This clearly implies that the pathways for the biosynthesis of AP-C12 to phospholipids, DAGs, and TAGs are markedly diminished by TC[35]. In other words, AP-C12 is distributed intracellularly after being converted to the corresponding fatty acyl-CoA in the same way as natural FAs, rather than being passively accumulated in the organelles as free FA.

Next, we monitored the distribution of AP-C12 metabolites using inhibitors of DGAT1 or DGAT2, which catalyzes the final acylation step in TAG synthesis. Although these enzymes are classified in the same family, they are functionally compartmentalized in FA metabolism, with the former preferentially using exogenously supplied FAs, whereas the latter is thought to be primarily responsible for the esterification of endogenously produced FAs[36]. We used T863 and PF-06424439 (PF), which are selective inhibitors for DGAT1 and DGAT2, respectively. When treated with T863, as is apparent in the merged image, the red dots corresponding to the LDs in the scatter plot disappeared, while the proportion of yellow dots significantly increased. This indicates that the inhibition of DGAT1 with T863 was quite effective for suppressing LD biogenesis with exogenous FAs including AP-C12 in the HepG2 cells[37,38]. Inhibition of TAG synthesis causes accumulation of unconsumed DAGs in the ER membrane, resulting in the enhancement in yellow, as observed in the 3T3-adipocytes. Subsequent phospholipid synthesis may possibly take place via conversion to phosphatidic acid (PA) by diacylglycerol kinase (DGK)[39]. On the other hand, although the inhibitory effect of PF on the lipogenesis using AP-C12 was relatively weak compared to that of T863, the number of red dots in the scatter plot meaningfully decreased than that for the control, indicating that DGAT2 also partially contributed to TAG synthesis from the exogenous FAs[38].

Finally, we applied this system to the evaluation of mitochondrial FA β-oxidation with etomoxir (ETO), a widely used inhibitor that irreversibly binds to the active site of CPT-1. Although both the images and the scatter plot of ETO-treated cells seem almost identical to those of the control (Supplementary Fig. 12), the former shows more nuclear LDs linked to nucleoplasmic reticulum (NR)[40,41] compared to the control (Supplementary Fig. 13). With CPT-1 inhibition, the excess FAs that have not been delivered to the mitochondrial matrix should remain in the cytosol as fatty acyl-CoAs, which are also re-esterified to produce TAGs and phospholipids[42]. The phenomenon observed with AP-C12 may be a visualization of the reported metabolic process in which excess phospholipids produced in the ER invaginate into the nucleus as the NR and form LDs in the NR lumen[40]. Consequently, no significant differences in the proportion of AP-C12 metabolites were found in the LDs, membranes, and aqueous environment, which may

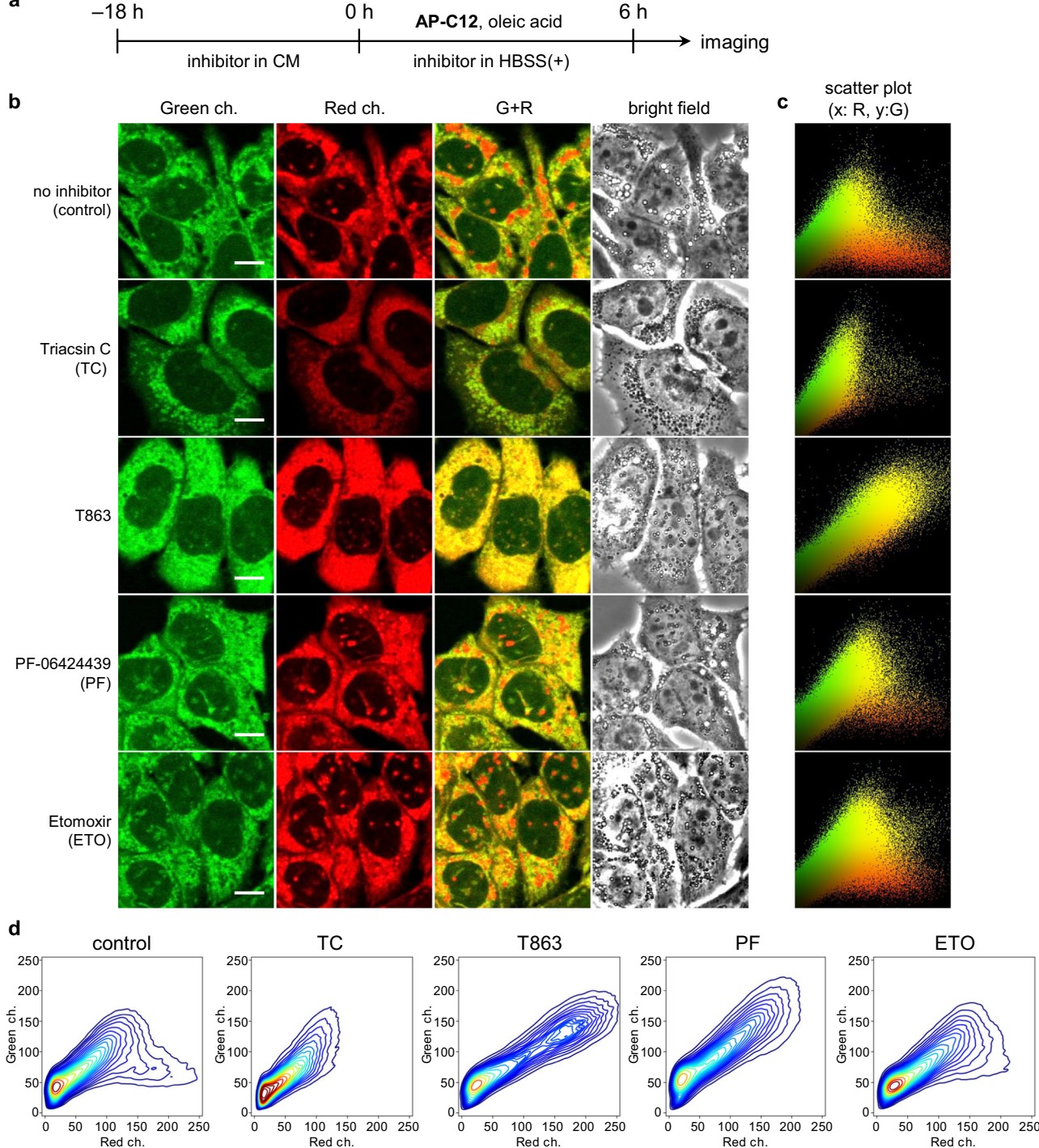

**Fig. 3 Distribution analysis of AP-C12 metabolites in HepG2 cells treated with fatty-acid-metabolism inhibitors under nutrition-starvation conditions.**
**a** Scheme of the preparation process for imaging: HepG2 cells were incubated in complete medium (CM) 18 h before the addition of each inhibitor. The medium was then replaced with HBSS containing 5 μM AP-C12, 0.5 mM oleic acid, and the indicated inhibitor. The cells were incubated for 6 h and then imaged. Inhibitors: triacsin C (TC), 5 μM; T863, 20 μM; PF-06424439 (PF), 10 μM; etomoxir (ETO), 10 μM. **b** Confocal images of HepG2 cells treated with the various inhibitors. From left to right: the images recorded in the green and red channels, the merged images of the two channels, and the bright-field images. The experiment was independently repeated twice with similar results. Scale bar, 10 μm. **c** Scatter plots of the signal intensities of the red (x axis) and green (y axis) channels observed in panel **b**. **d** Two-dimensional kernel density estimation (2D-KDE) contour plots for the distribution of the signal intensities corresponding to panel **b**.

account for the high levels of similarity between this image and that of the control. In addition to this, it has been reported that treatment with ETO induces oxidative stress and ATP depletion[43], as well as increased incorporation of palmitic acid into phospholipids and DAGs[44], which further complicates the analysis of the metabolic process of FAs. Although further investigations are needed to confirm the correlation of the imaging results with these biological events and clarify the

metabolic fate of AP-C12, given the successful observation of both NRs and LDs, AP-C12 would become a promising tool for a better understanding of the regulation mechanisms of FA metabolism.

Using the intensity distribution data obtained for the green and red channels, we conducted further analysis with two-dimensional kernel density estimation (2D-KDE), which is a non-parametric statistical method for estimating the probability density function (Fig. 3d). The region contours of the 2D-KDE for the scatter plots in Fig. 3c provide an intuitive representation of the distribution of AP-C12 metabolites, which allows classifying the metabolic patterns of FAs into three main patterns; type 1: Well metabolized to TAGs (e.g., control); type 2: Metabolized to DAGs and phospholipids (e.g., T863); type 3: Practically unmetabolized (e.g., TC). Thus, the effect of PF can be regarded as type 2, and that of ETO is likely to be intermediate between type 1 and type 2. This metabolic distribution analysis of FAs using AP-C12 and 2D-KDE can thus be expected to become a powerful method for facile drug screening without sophisticated devices.

**Lipid metabolism in autophagic cells.** The concentrations of free FAs in cells are homeostatically regulated for cell growth and phospholipid biosynthesis. Lipolysis of TAGs by ATGL on the surface of LDs and autophagic degradation of LDs (lipophagy) are the two principal pathways for the breakdown of LDs in hepatocellular lipid homeostasis[45–47]. Although the regulation mechanisms of FA homeostasis are still controversial[47,48], several experimental results have demonstrated that lipolysis is preferable to lipophagy for generating free FAs for β-oxidation in the mitochondrial matrix[11].

To understand how FAs are metabolized in nutrition-starved hepatocytes at the single-cell level, high magnification images of HepG2 cells were obtained using a TCS SP8 microscope (Leica) with a 100× objective lens (HC PL APO 100×/1.40 OIL CS2, Leica). The cells were treated with various reagents related to lipid metabolism in HBSS containing AP-C12, but not oleic acid, for 6 h, and the resulting images were analyzed. First, HBSS-starved cells were analyzed as the control. The merged image showed clear tubular mitochondria in yellow-green as well as LDs in red (control in Fig. 4a and Supplementary Fig. 14). In order to analyze the images more quantitatively, we prepared scatter plots of the signal intensities (red channel vs. green channel) in which the color represents the point density (Fig. 4b). The intensity ratio of the green and red channels (G/R) is sensitive to the environment polarity of the metabolic destinations of AP-C12; for example, mitochondria exhibit G/R ratios of 1.1–1.3, while G/R ratios of approximately 0.3 are observed for the lipid droplets (Fig. 4c). From the ratio image, the membrane structures including the ER were found to have G/R ratios of approximately 0.7. Therefore, we can intuitively understand the FA metabolism in starved cells from the density scatter plots in which the green, yellow, and red lines correspond to G/R ratios of 1.2, 0.7, and 0.3 (Fig. 4b).

We then analyzed the images of HepG2 cells after treatment with bafilomycin A1 (Baf-A1), 3-methyladenine (3-MA), diethyl-lumbelliferyl phosphate (DEUP), or rapamycin (Rapa) in the presence of AP-C12 for 6 h. Baf-A1 is a known autophagy inhibitor that blocks autophagosome–lysosome fusion, while 3-MA prevents autophagosome formation via the inhibition of class III phosphatidylinositol 3-kinases (PI-3K). By comparing the distribution patterns of the AP dyes in the two images, we were able to visualize how FAs are metabolized under the different autophagy inhibition conditions at the single-cell level.

Pretreatment with Baf-A1 induced an increase in the overall fluorescence intensity of both channels, whereas significant signal decreases were observed in the 3-MA-treated cells (Fig. 4d). The obvious signal increases in both channels in the merged image of the Baf-A1-treated cells as well as in the scatter plot are due to the dramatic increase in the vesicle-like membrane structures with a diameter of about 1 μm (Baf-A1 in Fig. 4a, b and Supplementary Fig. 15)[49,50]. In contrast, the signals from the LDs (G/R = 0.3) was efficiently suppressed by Baf-A1 (Baf-A1 in Fig. 4b)[51]. Separate experiments using GFP-LC3-expressing cells suggested that the vesicle-like structures induced by Baf-A1 treatment come from autophagosomal membranes, which drastically increased under the same culture conditions (Supplementary Fig. 16)[52]. On the other hand, no such distinctive membrane structures appeared when the cells were pretreated with 3-MA (3-MA in Fig. 4a and Supplementary Fig. 17)[11], which strongly supports the notion that AP-C12 is metabolically incorporated into the autophagosomal membrane. To corroborate this finding, we examined the metabolic distributions of the probe in $Atg5^{-/-}$ mouse embryonic fibroblasts (MEFs), which are unable to form autophagosomes[53]. Upon treatment with Baf-A1, the wild-type ($Atg5^{+/+}$ MEF) cells, such as the HepG2 cells, exhibited a large number of autophagosomes, while in $Atg5^{-/-}$ MEF cells, few of the vesicular morphologies that were thought to be derived from autophagosomes were observed (Fig. 4e).

In contrast to the autophagy-blocking conditions, the inhibition of lipase function with DEUP induced a considerable increase in the red channel only, resulting in a strong reddish color in the merged image (DEUP in Fig. 4a and Supplementary Fig. 18). The differences in the FA metabolic behavior are clearly seen in the density scatter plot, in which the number of pixels on the lines corresponding to a G/R of 0.7 and 0.3 increased markedly, whereas the signals originating from mitochondria almost disappeared (DEUP in Fig. 4b). Moreover, as opposed to the obvious decrease in the LD signal intensity in the Baf-A1- and 3-MA-treated cells compared to the control, DEUP-treated cells showed a high accumulation of TAGs containing the AP dye. These results suggest that the degradation of TAG is efficiently blocked by DEUP, and thus, TAG lipolysis in LDs is the dominant pathway to supply FAs for mitochondrial respiration in HepG2 cells[11,54]. The inhibition of lipolysis may result in the accumulation of de novo synthesized TAGs in the ER[54], which is assumed to enhance the red signal in the LDs and in the ER around the nucleus. This is also consistent with the experimental finding that when autophagosome formation is inhibited, the supply of TAGs into LDs is suppressed[11,52], thus reducing the fluorescence intensity of the AP dye. On the other hand, in primary mouse hepatocytes, it has been reported that lipophagy-derived FAs from LDs are extracellularly effluxed outside the cell under serum deprivation conditions, and then the FAs are transported back into the cell and supplied into mitochondria[55]. To avoid the autophagic degradation of LDs during pulsing, the cells were incubated with AP-C12 in DMEM containing 2% XerumFree (FBS replacement) for 1 h, and then chased in the absence or presence of Baf-A1, 3-MA or DEUP in HEPES-buffered HBSS (Supplementary Fig. 19). In the control and autophagy-inhibited cells, after the 6 h chase, significant decrease was observed, and negligible signals in fluorescence were detected both in the green and red channels, while in DEUP-treated cells, distinct fluorescence signals in LDs remained. AP-C12 is effectively oxidized in mitochondria and then released outside the cells, which seems to be shorter than the time required for induction of lipophagy; therefore, we cannot exclude the possibility of lipophagic degradation of LDs given the result of the ineffective inhibition by 3-MA. However,

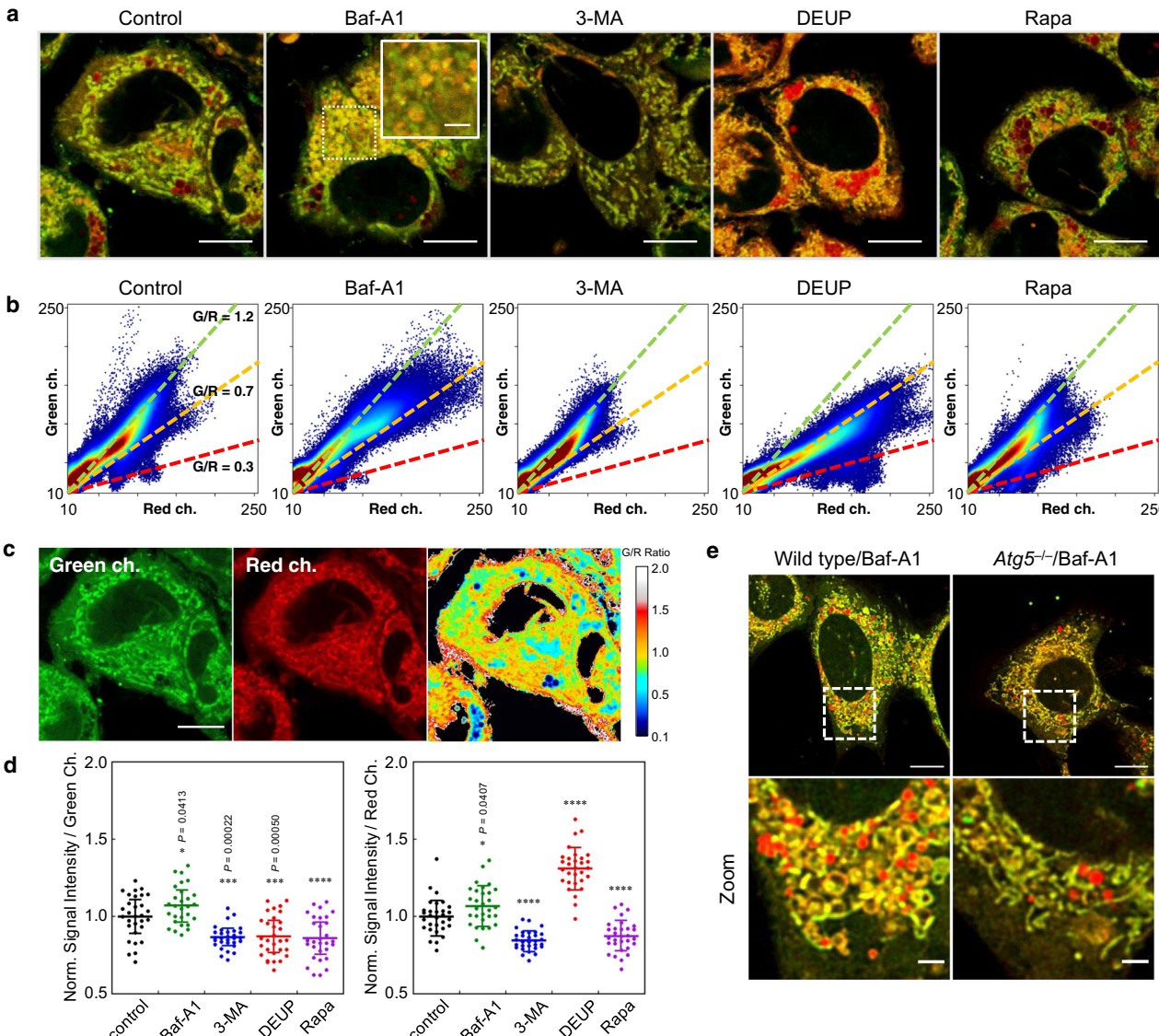

**Fig. 4 Image analysis of endogenous fatty-acid metabolism in autophagic cells. a** Merged images of HepG2 cells recorded in the green and red channels. The cells were incubated in HBSS containing 5 μM AP-C12 and the indicated reagents for 6 h. Reagents: 50 nM bafilomycin A1 (Baf-A1); 10 mM 3-methyladenine (3-MA); 100 μM diethylumbelliferyl phosphate (DEUP); 100 nM rapamycin (Rapa). Scale bar: 10 μm. The inset in the Baf-A1 panel shows a magnified image of the area indicated by the white dashed square. The experiment was independently repeated twice with similar results. **b** 2D histograms of the signal intensities of the red (x axis) and green (y axis) channels observed in panels **a**. Dashed lines in green, yellow, and red correspond to green/red channel (G/R) intensity ratios of 1.2, 0.7, and 0.3, respectively. **c** Image of the G/R channel ratio for HepG2 cells (control in panel **a**). Mitochondria and LDs are visualized as yellow-red and blue, respectively. Scale bar: 10 μm. **d** Intracellular signal intensities of the green (left) and red (right) channels upon incubation with the indicated reagents. The intensities are normalized to the average in the control. $n = 28$–30 cells. All statistical analyses are compared with the non-treatment group (control). *$P < 0.05$, ***$P < 0.001$, ****$P < 0.0001$ by unpaired two-tailed Student's t-test (data analysis tools in Microsoft Excel). The length of the error bars represents 1 standard deviation from the mean. **e** Merged images of wild type (left) and $Atg5^{-/-}$ (right) mouse embryonic fibroblasts incubated with AP-C12 and Baf-A1 in HBSS+ for 6 h. Images were deconvolved using Huygens software. Higher magnification images of the selected area (white dashed square) are shown at the bottom. The experiment was independently repeated twice with similar results. Scale bars: 10 μm (top) and 2 μm (bottom). Source data for this figure are provided as a Source Data file.

it is obvious that lipolysis is contributed to the degradation of relatively larger LDs under the HBSS-starved conditions. By accelerating autophagy beyond the rate that occurs under HBSS starvation using Rapa, many LDs increased in size and intensity were detected (Rapa in Fig. 4a, b and Supplementary Fig. 20)[52], revealing that autophagic process is also strongly related to the FA metabolism. Although it is still unclear how lipolysis and lipophagy are synergistically related to the degradation of LDs[45], the present results suggest that the inhibition of autophagy does indeed interfere with the supply of FAs to

mitochondria, albeit that this is not the predominant mode of supply; rather, LD lipolysis by ATGL is an essential mechanism of their degradation.

In summary, we have developed a fluorescent FA probe (AP-C12) that allows multicolor visualization of the FA metabolism in living cells based on the small size and excellent negative solvatochromic behavior of the AP dye. One important advantage of AP-C12 is that it allows for multicolor analysis of the metabolic distribution of FAs over various organelles with only one probe, whereas previously, the use a combination of multiple fluorescent

dyes with different absorption/emission properties had often been necessary. An analysis of the effects of inhibiting FA metabolism and autophagic processes using AP-C12 as the imaging probe implied that the degradation of LDs in starved cells might proceed, at least under the conditions applied in this study, via lipolysis rather than lipophagy. However, at current study, the mitochondrial localization mechanism of AP-C12, i.e., whether CPT-1 is required for the transportation of AP-C12 into the mitochondrial matrix, is not clear, and therefore further work is needed to validate this point. The solvatochromic features of AP-C12, together with its higher fluorescence quantum yields in aqueous media compared to standard probes, make it a valuable tool for further understanding intracellular lipid transportation and for drug discovery[56,57].

## Methods

**Synthesis of AP-C12.** AgSbF$_6$ (35.0 mg, 0.102 mmol) and potassium acetate (20.3 mg, 0.207 mmol) was added into an oven-dried Schlenk flask under an argon atmosphere. Then, [Cp*Co(CO)I$_2$] (23.8 mg, 50 μmol), benzo[h]quinoline (91.6 mg, 0.511 mmol), and bis(12-tert-butoxy-12-oxododecyl)-2-diazomalonate (478 mg, 0.749 mmol) in 2,2,2-trifluoroethanol (5 mL) was added. The resulting suspension was stirred at 80 °C for 16 h. The mixture was concentrated under reduced pressure and subjected to silica gel column chromatography (dichloromethane/acetone = 20:1 then 4:1). Further purification by preparative gel permeation chromatography (GPC) with chloroform afforded 12-tert-butoxyl-12-oxododecyl 3a-azapyren-4-one-5-carboxylate (**1b**) as a red solid (80.1 mg, 0.155 mmol, 30%). A solution of **1b** (54.6 mg, 0.106 mmol) in trifluoroacetic acid (2 mL) and dichloromethane (2 mL) was stirred at ambient temperature for 12 h. Water was added, and the organic layer was separated. Aqueous layer was extracted with dichloromethane, and combined organic extracts were washed with brine, dried over sodium sulfate, filtered, and concentrated under reduced pressure. The obtained crude product was suspended in small amounts of dichloromethane and filtered to give red solids. The obtained solids were dried in an oven (100 °C) to afford **AP-C12** (24.9 mg, 53.9 μmol, 51%) as a red solid. Full experiment details can be found in the Supplementary Information.

**Photophysical properties.** UV-visible absorption spectra were recorded on an Agilent 8453 UV-visible spectroscopy system with a resolution of 1 nm. Steady-state fluorescence spectra of sample solutions were recorded on a FluoroMax-4 spectrometer (HORIBA) with a resolution of 1 nm. Spectroscopic grade organic solvents and 0.1 M sodium phosphate buffer (pH 7.4) were used for the measurement. The sample solutions of ca. 10$^{-5}$ M in a 1 cm square quartz cuvette were measured. For the fluorescence measurement, the sample solutions were excited at the absorption maximum wavelengths for each compound. Absolute fluorescence quantum yields were determined with a Quantaurus-QY C11347-02 (Hamamatsu Photonics) calibrated integrating sphere system equipped with multichannel spectrometer (PMA-11).

**Theoretical calculations.** Geometry optimizations were performed using the Gaussian 09 program at the B3LYP/6-31+G(d) level of theory including the effect of solvents by PCM model. For estimation of the NICS (nucleus-independent chemical shift) values, GIAO calculations were performed at the HF/6-31+G(d) level using the optimized geometry.

**AP-C12 in model systems.** A stock solution of AP-C12 in chloroform was diluted with soybean oil (FUJIFILM Wako), followed by heating under vacuum to remove chloroform. Large unilamellar vesicles (LUVs) containing AP-C12 were prepared as follows. Dry lipid film was prepared from chloroform solution of 1,2-dioleoy-sn-glycero-3-phosphocholine (DOPC) and AP-C12 in 200/1 molar ratio. The film was hydrated with phosphate buffer (pH 7.4) to make the final concentration of the lipid as 1 mM, and LUVs were prepared with the aid of sonication. Obtained vesicle solution was extruded through a 100 nm pore polycarbonate filter (10 times) using Avanti Mini-Extruder. LUVs solution was characterized by dynamic light scattering (DLS). Regarding an aqueous solution, a stock solution of AP-C12 in dimethyl sulfoxide (DMSO) was diluted with phosphate buffer (pH 7.4, 1% DMSO as a co-solvent).

**Cell culture.** 3T3-L1 cells (IFO50416) were purchased from JCRB Cell Bank (Japan). HepG2 (RCB1648), Atg5$^{+/+}$ MEF (RCB2710), and Atg5$^{-/-}$ MEF (RCB2711) cells were obtained from RIKEN BRC Cell Bank (Japan). The cells were cultured in complete medium (CM; DMEM containing 4 mM glutamine (FUJIFILM Wako) and 10% FBS (Biosera)) with 1% antibiotic/antimitotic (AA; penicillin, streptomycin, and amphotericin B) at 37 °C in a 5% CO$_2$/95% air incubator. One day before imaging, cells (5 × 10$^4$ cells/mL) were transferred on a glass-bottom dish (Matsunami Glass) and cultured. To initiate 3T3-L1

adipocyte differentiation, 2 days after cells reached 100% confluence, the medium was replaced with CM containing hormone cocktail (10 μg/mL insulin, 2.5 μM dexamethasone and 0.5 mM 3-isobutyl-methylxanthine). After incubated for 2 days, the medium was changed to the maintenance medium for adipocyte (CM with 10 μg/mL insulin) and the medium was exchanged every 2 days. To make HepG2 cells form large LDs, HepG2 cells were incubated with CM containing 1 mM fatty acids (oleate/palmitate 2:1) complexed and incubated for 6 h before imaging.

**Cell viability assays.** The cell viability after treatment with AP-C12 was measured by the MTT (MTT: 3-(4,5-dimethylthiazol-2-yl)-2,5-diphenyltetrazolium bromide) assay. HepG2 cells were seeded into a flat-bottomed 96-well plate (Nunc) and incubated in CM at 37 °C in a CO$_2$ incubator for 24 h. The medium was then replaced with DMEM containing various concentrations of AP-C12 (0, 1, 5, 10, and 20 μM) and 0.1% DMSO. After incubation for 24 h, MTT reagent (final concentration, 0.5 mg/mL, Dojindo) was added to each well, and the plates were incubated for another 4 h in a CO$_2$ incubator. Excess MTT tetrazolim solution was then removed. After the formazan crystals were solubilized in DMSO (200 μL/well) for 30 min at room temperature, the absorbance of each well at 535 nm was measured by SpectraMax i3 (Molecular Devices). Cell viability was determined by the following formula (1).

$$\text{Cell viability } (\%) = (A_{sample} - A_{blank})/(A_{control} - A_{blank}) \times 100 \qquad (1)$$

A$_{sample}$ is the average of absorbance in wells of a certain reagent concentration. A$_{control}$ is the average of absorbance in wells only with cells. A$_{blank}$ is the average of absorbance in wells only with solvents.

**TCL analysis of AP-C12 metabolites.** HepG2 cells cultured in 10 cm dishes were treated with 500 μM of oleic acid in CM and incubated in a 5% CO$_2$/95% air incubator at 37 °C for 1 day. The cells were washed with HBSS+ three times and incubated with 5 μM of AP-C12 in HEPES-buffered HBSS+ for 6 h. After removal of the dye-containing medium, the cells were harvested with trypsin and centrifuged at 100 × g for 5 min, then the supernatant was decanted. 3 mL of PBS was added and centrifuged again for 5 min. After repeating this procedure once again, 1 mL of chloroform and 0.5 mL of methanol were added and sonicated to disrupt cells. Then, 0.5 mL of chloroform and 2 mL of water were added, followed by centrifugation at 5000 rpm for 5 min. After the aqueous layer was removed, 2 mL of water was added and centrifuged again at 5000 rpm for 5 min. Then, organic layer was collected. Separation of lipids was performed by developing the TLC plates in a solvent system of chloroform/methanol, 9:1 (vol/vol). Fluorescent lipids were visualized with a Typhoon FLA9500 (GE HealthCare).

**MS analysis.** HepG2 cells were incubated with HBSS+ containing 5 μM of AP-C12 or BODIPY 558/568 C12 (Invitrogen) for 1 h. After collecting the medium, cells were detached using trypsin and centrifuged at 100 × g for 5 min. The supernatant was decanted, 3 mL of PBS was added, and the cells were centrifuged again for 5 min. After repeating this procedure once again, 1 mL of chloroform was added and sonicated to disrupt cells. Then, 1 mL of chloroform and 2 mL of water were added, followed by centrifugation at 14,000 rpm for 5 min. The aqueous and the organic layers were collected separately. The same extraction was conducted on collected medium. 1 mL of each extract was concentrated by centrifugation, and the residue was dissolved with 10 μL of methanol. After dilution with 20 μL of 0.1% formic acid, centrifugation at 15,000 × g was conducted for 5 min. 1 μL of the supernatants were analyzed by LC-MS, combination of Dionex Ultimate 3000 HPLC and EXACTIVE Plus mass spectrometer (ThermoFisher). As an analytical column, Accucore RP-MS column (2.6 μm, 2.1 × 50 mm, ThermoFisher) was used. Column temperature; 60 °C. Flow rate; 200 μL/min. Mobile phase A; 0.1% aqueous formic acid. Mobile phase B; acetonitrile. Gradient; 5% B for 0–2 min, 5% B to 90% B for 2–4 min, and finally 80% B for 4–10 min. Samples were detected in ESI positive ion mode. Relative quantification was conducted with a calibration curve. Xcalibur ver. 2.2 software was used for the analysis. For the detection of degradation product of AP-C12, the in-source CID was set to 50, and an extracted ion chromatogram of the dye ion (C$_{16}$H$_8$NO$_2^+$, 246.0550) whose fatty acid was released was prepared.

**Confocal microscopy.** Olympus FV10i-DOC or a Leica TCS SP8 STED 3X system including an inverted DMI6000 CS microscope equipped with a tunable (470–670 nm) pulsed white light laser (WLL; repetition rate of 78 MHz) was used. For high magnification observation, HC PL APO CS2 100×/1.40 oil objective was used. Typically, the cells were incubated with 5 μM AP-C12 in DMEM (phenol red free, 1% AA) or HEPES-buffered HBSS+ (HBSS with Mg$^{2+}$ and Ca$^{2+}$) at 37 °C in a 5% CO$_2$ atmosphere, and imaged without washing process. Images of cells stained with AP-C12 were acquired with green channel (λ$_{ex}$ = 473 nm, λ$_{em}$ = 490–540 nm) and red channel (λ$_{ex}$ = 559 nm, λ$_{em}$ = 570–620 nm). Cells stained with MitoTracker Deep Red were excited with a 635 nm and the emission was detected at 660–710 nm.

**Spectral imaging of 3T3-L1 adipocytes**. Before staining with AP-C12, 3T3-adipocytes were treated with 1.2 mM of 2-bromooctanoate in CM for 30 min. Then, 3T3-L1 cells were incubated with DMEM (phenol red free, 1% AA) containing 5 μM of AP-C12 for 1 h. Cell images and fluorescent spectra with a resolution of 8 nm were recorded on Zeiss LSM780-DUO-NLO with 32-channel spectral GaAsP detector and a Plan-Apochromat 63x/1.40 oil immersion objective lens ($\lambda_{ex}$ = 458 nm, $\lambda_{em}$ = 468–689 nm).

**Imaging of metabolic pathways of AP-C12**. HepG2 cells were incubated with each inhibitor in CM for 18 h (5 μM of triacsin C, 20 μM of T863, 10 μM of PF-06424439 and 10 μM of etomoxir). After washing with HBSS+ three times, the cells were incubated with 5 μM of AP-C12, 500 μM of oleic acid and each inhibitor (the same concentration) in HEPES-buffered HBSS+ for 6 h. The cells were directly imaged using the green and red channels without washing process. For counting the number of LDs, HepG2 cells treated with or without 10 μM of etomoxir for 18 h and stained with 1 μM of LipiDye II (Funakoshi) in HEPES-buffered HBSS+ for 3 h. SiR-DNA (Cytoskeleton, Inc.) was used for imaging cell nucleus. Counting the number of LDs and analyzing their size were performed using ImageJ. Fluorescence intensity below the threshold was ignored. $\lambda_{ex}$ = 473 nm, $\lambda_{em}$ = 490–540 nm. Images were analyzed for ca. 30 cells for statistics. 2D-Kernel density estimation (2D-KDE) was created using Matplotlib in Python. The images are representative of more than 10 cells obtained from two independent experiments.

**Imaging of AP-C12 in autophagic cells (HepG2 cells)**. After HepG2 cells were washed with HBSS+ three times, the cells were incubated with 5 μM of AP-C12 and the following reagent in HEPES-buffered HBSS+ for 6 h; 50 nM bafilomycin A1 (Baf-A1), 10 mM 3-methyladenine (3-MA), 100 μM diethylumbelliferyl phosphate (DEUP), or 100 nM rapamycin (Rapa). MEFs were imaged after incubation in HEPES-buffered HBSS+ containing 5 μM of AP-C12 and 50 nM of Baf-A1 for 6 h. For pulse-chase experiments, HepG2 cells were pulsed with 5 μM of AP-C12 in DMEM supplemented with XerumFree (TNC BIO) for 1 h. The cells were rinsed with HBSS+ three times and chased in HEPES-buffered HBSS+ containing 50 nM Baf-A1, 10 mM 3-MA, 100 μM DEUP, or 100 nM Rapa for 6 h. Confocal images were recorded in the green and red channels without washing the cells. The images are representative of more than 10 cells obtained from two independent experiments.

Cells were transfected with GFP-LC3 (pCMV-GFP-LC3 expression vector, Cell Biolabs, Inc.) using Lipofectamine 3000 (Thermo Fisher Scientific) according to the manufacturer's instructions. Fluorescence of GFP was detected with $\lambda_{ex}$ = 473 nm and $\lambda_{em}$ = 490–540 nm.

**Statistical analysis and reproducibility**. Each cell imaging experiment reported in this study was independently repeated at least twice with similar results. The number of total analyzed cells and the statistical test used are indicated in figures legends.

**Reporting summary**. Further information on research design is available in the Nature Research Reporting Summary linked to this article.

## Data availability
All data generated in this study are provided in the Supplementary Information and Source Data file.

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

## Acknowledgements

We would like to thank the following people for their technical assistance or sharing materials: Y. Ohsaki (Sapporo Medical University) for discussing the fatty acid metabolism. This work was supported by JST PRESTO JPMJPR16F5 (MT), JSPS KAKENHI grants 19H02849 (MT) and JP16H06280 (Advanced Bioimging Support). KK thanks the JSPS for a Research Fellowship for Young Scientists. This work is partially supported by Nagoya University Research Fund. ITbM is supported by the World Premier International Research Center (WPI) Initiative (Japan).

## Author contributions

M.T., S.Y., and F.G. conceived and supervised the project. K.K. and H.O. performed most of the experiments including the organic synthesis, spectroscopy measurements, and cell-imaging experiments. S.G., J.H.K., T.G., and F.G. led the molecular design and organic synthesis of the probes. K.K. conducted the mass analysis of the metabolic compounds. Y.S. contributed to the cell-imaging. K.K., H.O., S.G., F.G., S.Y., and M.T. wrote the paper. All authors edited the paper.

## Competing interests

The authors declare no competing interests.
