## [Peer Review File · Nature Communications]

REVIEWER COMMENTS

Reviewer #1 (Remarks to the Author):

The manuscript by Yamaguchi, Glorius and co-workers is focused on the discovery of the entirely new fluorescence probe for fatty acids metabolites. The probe is based on the new skeleton (azapyrene) discovered only in 2015 and very rarely explored. Authors recognized its excellent photophysical properties especially the combination of large fluorescence quantum yield (regardless the solvent polarity) and negative solvatochromism. One practical advantage is that the synthesis is straightforward and comprises of only two steps from commercially available materials. More importantly however the functional properties are excellent. Among many advantages one which is essentially unique is that this fluorophore does not interfere with the beta-oxidation process (which leads to the fact that old probes are not recognized as substrates in metabolic processes). Having this probe authors were able for the first time:

1. Perform multicolor analysis of the metabolite distribution of fatty acids over various organelles with only one probe.
 2. Prove that lipid droplets degradation in starved cells proceeds via lipolysis rather than via lipophagy.
- The fundamental advantage of the new fl. probe is that it possesses high sensitivity to local environment polarity of the organelles in which FA metabolites are present. This represents the ground-breaking achievement and will be likely followed by others working in this field. This is especially true for studies of mitochondrial metabolism. The manuscript is excellently written. I have not spotted any issues with methodology in fluorescence microscopy experiments.

Minor point:

Line 98 – replace ‘benzo[h]quinolone’ with ‘benzo[h]quinoline’

Reviewer #2 (Remarks to the Author):

In this manuscript, Kajiwara and colleagues develop a new fluorescent probe, azapyrene (AP)-C12, to follow the trafficking of fatty acids in live cells. The AP dye changes its absorption and emission features depending on the microenvironment of the organelle to which it is transported (e.g., lipid droplets vs. mitochondria). The authors show that AP-C12 is oxidized more efficiently than BODIPY558/568-C12, the most widely used fluorescent fatty acid analogue. They propose that AP-C12 can be used as a spectral reporter of fatty acid fate within living cells. This idea is quite innovative, and the data presented are of high quality. However, I have several concerns.

The authors characterize the oxidation of AP-C12 using LC/MS, showing that it is oxidized more efficiently than BODIPY558/568. However, they do not directly test the ability of AP-C12 to be incorporated into phospholipids or neutral lipids. Nor do they compare the kinetics of metabolism to that of a radiolabelled fatty acid, which is necessary for a complete characterization of this tool. Finally, the authors do not use the tool to discover any novel biology. In Figure 4, they perform experiments where they starve HepG2 cells in the presence of AP-C12 and inhibitors of autophagy or lipolysis,

concluding that lipolysis is the major pathway for delivering fatty acids to mitochondria. However, this was already shown in other cell types using BODIPY558/568-C12 (Rambold et al., Dev. Cell 2015). In hepatocytes, an alternative trafficking route has been proposed. It was proposed that starvation induces lipophagy (autophagy of lipid droplets), followed by fusion of lysosomes with the plasma membrane; this releases fatty acids outside the cell, which are then transported back into the cell and routed to mitochondria (Cui et al., Autophagy 2021). However, with their experimental design of adding AP-C12 during starvation, rather than pulsing it into lipid droplets prior to starvation, the authors do not directly test this model. Thus, these experiments do not significantly advance our understanding of intracellular fatty acid trafficking, and may actually confuse the field by having a different design from the pulse-chase approach used with BODIPY558/568-C12 in previous studies.

In summary, I believe the paper in its current form is better suited to a journal focused on methods development. To warrant publication in Nature Communications, the authors would need to demonstrate that AP-C12 can in fact be used to reveal new biological insight.

Major Points

1. In Supp. Figs. S5-S9, the authors compare the metabolism of AP-C12 to that of BODIPY 558/568-C12. They show that AP-C12 is more readily oxidized (>20% oxidized?) than B558/568-C12 (<3% oxidized). However, it would be ideal to compare the metabolism of AP-C12 with that of a radioactively-labelled long chain fatty acid, that is not conjugated to a fluorophore. Such a comparison should include a measure of how much AP-C12 is incorporated into phospholipids and neutral lipids, to better characterize the metabolism of AP-C12 relative to a “natural” fatty acid.
2. In Fig. 3, the authors use inhibitors to test whether the spectral properties of AP-C12 can be used to distinguish its subcellular localization under different conditions. They argue that AP-C12 that fluoresces in the green channel is in the mitochondrial matrix, that AP-C12 that fluoresces in the red channel is in lipid droplets, and that AP-C12 that fluoresces in both (appears yellow) is in membranes. However, they do not observe decreased green signal when cells are treated with etomoxir, which inhibits transport of fatty acids into mitochondria. This calls the authors interpretation into question. More work is needed to determine the localization and metabolic fate (e.g., oxidation, incorporation into phospholipids or neutral lipids) of AP-C12 with different spectral properties.
3. In Fig. 4, the authors starve HepG2 cells in the presence of AP-C12 and different inhibitors. Inhibiting lipolysis causes buildup of AP-C12 in lipid droplets, while inhibiting lipophagy does not. However, the results are a bit difficult to interpret, because the AP-C12 is added during starvation. The figure is called “image analysis of endogenous fatty acid metabolism in autophagic cells”, however the AP-C12 is being added exogenously during starvation. If lipophagy of endogenous neutral lipids occurs early during starvation, this would not be detected due to the study design. I believe it would make more sense to pulse the AP-C12 into the cells prior to starvation, and then starve in the presence of the inhibitors.

Minor Points

Line 16: FAs are not the sole precursors of most lipids. What about glycerol? Sphingosine?

Line 29: It is not clear what is meant by invasive. Why is imaging mass spectrometry or alkyne-labelled FAs more “invasive” than fluorescent probes? Do the authors mean those other techniques are not compatible with live cell-imaging?

Reviewer #3 (Remarks to the Author):

The manuscript by Kajiwara et al describes a fatty acid derivative (AP-C12) of a fluorescent dye presenting negative solvatochromism. It is shown that this probe can be involved in different metabolic pathways that can direct it to different organelles, where it changes the color because of differences in local polarity. A variety of inhibitors of enzymes were used to block some metabolic pathways of lipids, which allowed visualizing the response of the probe. Overall, the manuscript is well written, the idea to combine solvatochromism of a probe with its metabolic transformation is rather novel and the results obtained are interesting. The developed probe may find a number of applications to monitor distribution and metabolism of fatty acids and to screen relevant drugs. The technical quality of the work is good, although some additional quantification analysis should be provided. Moreover, the manuscript lacks appropriate positioning of the developed probe with respect to existing solvatochromic dyes and some additional experimental data should be provided to support the claims. Therefore, I recommend this manuscript for publication in Nature Communications after major revisions. The detailed comments are provided below.

Major issues:

1) There has been an intensive research on the design and application of solvatochromic dyes to visualize variations at the level of lipid membranes, organelles and lipid droplets. The examples include Nile Red and its derivatives (DOI: 10.1021/jacs.7b03846; 10.1021/jacs.0c10972; 10.1073/pnas.2016897118;); push-pull pyrene (DOI: 10.1038/srep18870; 10.1021/acs.analchem.0c00023); and Laurdan (DOI: 10.1038/nprot.2011.419; 10.12688/f1000research.11577.2; doi.org/10.1073/pnas.1118288109), which present positive solvatochromism. All these probes revealed fine differences in the local polarity within the organelles (in line with those shown in this work) as well as showed sensitivity the cellular processes. In particular, the effects of starvation, oxidative stress and autophagy have been already studied by the solvatochromic probes (e.g. DOI: 10.1021/acs.jpcllett.9b00668; 10.1039/D1AN00667C). All these works (and other relevant ones) should be carefully mentioned in the Introduction in order to properly position the present solvatochromic probe with respect to the current state of the art. Moreover, the authors should make comparison of the new probe with already reported solvatochromic probes in the Conclusions section or end of Results and Discussion. Currently, this comparison is missing in the manuscript.

2) Page 7: “Incubation for a further 30 min resulted in an increase in the fluorescence signals from LDs in the red channel, while relatively weak fluorescence with no distinct features was observed in the green channel (Figure 2a, middle). This suggests that AP-C12 was taken up by the cells as an exogenous FA and readily converted into the corresponding TAGs in a similar manner to natural FAs, and that the TAGs subsequently accumulated in LDs.” This is actually the central claim of the present work, but there is no direct experimental support for it. The authors made an experiment with an inhibitor of DTAG to support indirectly this claim. However, to provide a solid support that the probe AP-C12 is really

converted into TAG and thus accumulates inside LDs, the authors should make LC-MS spectral analysis from the cell extract with chloroform, similarly to that performed for analysis of the beta-oxidation products.

3) Data analysis should be improved to provide more quantitative description of the solvatochromic response of the probes. At the moment the probes response is analyzed in form of ratiometric images and scatter plots, which do not really provide the values. The attempt to quantify the data is given in Figure 4d, but in this case, red and green channels are presented separately. However, the real advantage of solvatochromic dyes is a possibility to perform ratiometric data analysis, which are independent of local dye concentration. Therefore, the authors should provide systematically the values of green to red ratio and this should be done for the whole cell and some regions of interest (e.g. lipid droplets, ER and mitochondria, when they are well identified).

Minor points:

1) Page 4: “negative solvatochromic properties” in the subtitle should be changed to “negative solvatochromism”.

2) Page 16: “polarity environment” should be “environment polarity”.

3) Page 19: Last paragraph is not clear because in the first sentence the authors mention “accelerating autophagy” and then, in next one, they talk about “inhibition of autophagy”. These two sequences should be better connected to improve clarity.

4) Conclusions section “fluorescent yields” should be “fluorescence quantum yields”. Moreover, it is not clear what is meant by “standard probes”. The latter issue is connected with the problem mentioned in point 1 of the major issues.

Reviewer #1 (Remarks to the Author):

The manuscript by Yamaguchi, Glorius and co-workers is focused on the discovery of the entirely new fluorescence probe for fatty acids metabolites. The probe is based on the new skeleton (azapyrene) discovered only in 2015 and very rarely explored. Authors recognized its excellent photophysical properties especially the combination of large fluorescence quantum yield (regardless the solvent polarity) and negative solvatochromism. One practical advantage is that the synthesis is straightforward and comprises of only two steps from commercially available materials. More importantly however the functional properties are excellent. Among many advantages one which is essentially unique is that this fluorophore does not interfere with the beta-oxidation process (which leads to the fact that old probes are not recognized as substrates in metabolic processes). Having this probe authors were able for the first time:

1. Perform multicolor analysis of the metabolite distribution of fatty acids over various organelles with only one probe.
2. Prove that lipid droplets degradation in starved cells proceeds via lipolysis rather than via lipophagy.

The fundamental advantage of the new fl. probe is that it possesses high sensitivity to local environment polarity of the organelles in which FA metabolites are present. This represents the ground-breaking achievement and will be likely followed by others working in this field. This is especially true for studies of mitochondrial metabolism. The manuscript is excellently written. I have not spotted any issues with methodology in fluorescence microscopy experiments.

Thank you very much for high evaluation of our works.

Minor point:

Line 98 – replace ‘benzo[h]quinolone’ with ‘benzo[h]quinoline’

Re: We have corrected it as pointed out.

Reviewer #2 (Remarks to the Author):

In this manuscript, Kajiwara and colleagues develop a new fluorescent probe, azapyrene (AP)-C12, to follow the trafficking of fatty acids in live cells. The AP dye changes its absorption and emission features depending on the microenvironment of the organelle to which it is transported (e.g., lipid droplets vs. mitochondria). The authors show that AP-C12 is oxidized more efficiently than BODIPY558/568-C12, the most widely used fluorescent fatty acid analogue. They propose that AP-C12 can be used as a spectral reporter of fatty acid fate within living cells. This idea is quite innovative, and the data presented are of high quality. However, I have several concerns.

The authors characterize the oxidation of AP-C12 using LC/MS, showing that it is oxidized more efficiently than BODIPY558/568. However, they do not directly test the ability of AP-C12 to be incorporated into phospholipids or neutral lipids. Nor do they compare the kinetics of metabolism to that of a radiolabelled fatty acid, which is necessary for a complete characterization of this tool. Finally, the authors do not use the tool to discover any novel biology. In Figure 4, they perform experiments where they starve HepG2 cells in the presence of AP-C12 and inhibitors of autophagy or lipolysis, concluding that lipolysis is the major pathway for delivering fatty acids to mitochondria. However, this was already shown in other cell types using BODIPY558/568-C12 (Rambold et al., *Dev. Cell* 2015). In hepatocytes, an alternative trafficking route has been proposed. It was proposed that starvation induces lipophagy (autophagy of lipid droplets), followed by fusion of lysosomes with the plasma membrane; this releases fatty acids outside the cell, which are then transported back into the cell and routed to mitochondria (Cui et al., *Autophagy* 2021). However, with their experimental design of adding AP-C12 during starvation, rather than pulsing it into lipid droplets prior to starvation, the authors do not directly test this model. Thus, these experiments do not significantly advance our understanding of intracellular fatty acid trafficking, and may actually confuse the field by having a different design from the pulse-chase approach used with BODIPY558/568-C12 in previous studies.

In summary, I believe the paper in its current form is better suited to a journal focused on methods development. To warrant publication in *Nature Communications*, the authors would need to demonstrate that AP-C12 can in fact be used to reveal new biological insight.

Re: The biggest difference of AP-C12 from natural FAs is that the former behaves like FA in cells, but β -oxidation products with a shorter carbon chain, such as AP-C6, is not recognized as FA because of the strong influence of the AP fluorophore. We understand that there are certain limitations caused by the modification with the fluorophore, and that further studies are needed to assess its fidelity as FA for broader applications. However, we believe that our newly developed fluorescent FA and its analytical method will be a powerful tool for research on the FA metabolism. We would like to answer the following comments point-by-point.

Major Points

1. In Supp. Figs. S5-S9, the authors compare the metabolism of AP-C12 to that of BODIPY 558/568-C12. They show that AP-C12 is more readily oxidized (>20% oxidized?) than B558/568-C12 (<3% oxidized). However, it would be ideal to compare the metabolism of AP-C12 with that of a radioactively-labelled long chain fatty acid, that is not conjugated to a fluorophore. Such a comparison should include a measure of how much AP-C12 is incorporated into phospholipids and neutral lipids, to better characterize the metabolism of AP-C12 relative to a “natural” fatty acid.

Re: Thanks for the suggestion. We agree that the comparison of the metabolism of AP-C12 with that of a radiolabeled FA will give much information about the metabolic efficiency of AP-C12, which is a critical experiment to assess the fidelity of the probe as FA in cells. However, because of the regulations on the use of radioactive material, it is difficult for us to perform this experiment. Moreover, as observed by the MS analysis, the β -oxidation of AP-C12 in mitochondria does not proceed to the end and the products with a shortened carbon chain (*e.g.* less than six) are effluxed from cells (shown in Figure S6); therefore, a direct comparison of metabolic efficiency may not be reasonable in this case. Instead, we performed a chromatographic analysis of the AP-C12 metabolites to confirm that AP-C12 is incorporated into phospholipids or neutral lipids.

After treatment with AP-C12 for 6 h, trypsinized HepG2 cells were harvested and homogenized in $\text{CHCl}_3/\text{MeOH}$. The sample separated on a TLC plate was imaged by a fluorescence scanner (Typhoon FLA9500, GE HealthCare). In addition of original AP-C12, many spots corresponding to the metabolic products were newly observed. Based on the retention time of the authentic sample, AP-C6, as well as the reported results using

radioactive FAs, the spots seen just below AP-C12 may correspond to the β -oxidation products, whereas the spots appearing above AP-C12 could be attributed to TAGs and cholesterol ester containing at least one AP moiety. The lower spots with $R_f = 0.15$ and the origin may be from DAGs and phospholipids, respectively.

Regarding this point, we have added the TLC image to Figure 2d and the explanations to the main text on page 10 as follows. The detail of the experimental procedure was shown in the Supporting Information.

“To confirm that AP-C12 is metabolically distributed in the cells, after HepG2 cells were incubated with the probe in HBSS+ for 6 h, the cell lysate was separated on a thin-layer chromatography (TLC) plate, and then imaged using a fluorescence scanner. In addition to the original spot of AP-C12 ($R_f = 0.48$), many spots corresponding to the metabolic products were newly observed (Figure 2d). Based on the retention time of the authentic sample, AP-C6, and the reported results using radioactive FAs³¹, the spots seen just below AP-C12 are ascribed to the β -oxidation products, whereas the spots appearing above AP-C12 could be attributed to TAGs and cholesterol ester containing at least one AP moiety. Moreover, the lower spot with $R_f = 0.15$ and the origin may be from DAGs and phospholipids, respectively. This result supports that the cell images observed with AP-C12 are from the metabolized probes localized in the respective organelle.”

31. Bartz, R. *et al.* Lipidomics reveals that adiposomes store ether lipids and mediate phospholipid traffic. *J. Lipid Res.* **48**, 837–847 (2007).

2. In Fig. 3, the authors use inhibitors to test whether the spectral properties of AP-C12 can be used to distinguish its subcellular localization under different conditions. They argue that AP-C12 that fluoresces in the green channel is in the mitochondrial matrix, that AP-C12 that fluoresces in the red channel is in lipid droplets, and that AP-C12 that

fluoresces in both (appears yellow) is in membranes. However, they do not observe decreased green signal when cells are treated with etomoxir, which inhibits transport of fatty acids into mitochondria. This calls the authors interpretation into question. More work is needed to determine the localization and metabolic fate (e.g., oxidation, incorporation into phospholipids or neutral lipids) of AP-C12 with different spectral properties.

Re: It has been reported that etomoxir not only inhibits CPT-1 mediated pathway of free FAs, but also affects other cellular events. Xu et al. reported that, in cardiomyocyte H9c2 cells, etomoxir treatment results in an increase of FA incorporation into phosphatidylcholine, phosphatidylethanolamine, and diacylglycerol (DAG), but a decrease of FA uptake into triacylglycerol (TAG) (*J. Lipid Res.* 2003, ref 44). This is consistent with the experimental results obtained in Figure 3, *i.e.*, the increase in green and red signals corresponding to the AP dye in the lipid membrane, although it is not certain that external FAs undergo the same metabolic processes in the presence of etomoxir in these different cell types. Regarding this point, the following sentences have been added to the main text on pages 15-16.

“In addition to this, it has been reported that treatment with ETO induces oxidative stress and ATP depletion⁴³, as well as increased incorporation of palmitic acid into phospholipids and DAGs⁴⁴, which further complicates the analysis of the metabolic process of FAs. Although further investigations are needed to confirm the correlation of the imaging results with these biological events and clarify the metabolic fate of AP-C12, given the successful observation of both NRs and LDs, AP-C12 would become a promising tool for a better understanding of the regulation mechanisms of FA metabolism.”

43. Amen, T. & Kaganovich, D. Stress granules inhibit fatty acid oxidation by modulating mitochondrial permeability. *Cell Rep.* **35**, 109237 (2021).

44. Xu, F. Y., Taylor, W. A., Hurd, J. A. & Hatch, G. M. Etomoxir mediates differential metabolic channeling of fatty acid and glycerol precursors into cardiolipin in H9c2 cells. *J. Lipid Res.* **44**, 415–423 (2003).

3. In Fig. 4, the authors starve HepG2 cells in the presence of AP-C12 and different inhibitors. Inhibiting lipolysis causes buildup of AP-C12 in lipid droplets, while

inhibiting lipophagy does not. However, the results are a bit difficult to interpret, because the AP-C12 is added during starvation. The figure is called “image analysis of endogenous fatty acid metabolism in autophagic cells”, however the AP-C12 is being added exogenously during starvation. If lipophagy of endogenous neutral lipids occurs early during starvation, this would not be detected due to the study design. I believe it would make more sense to pulse the AP-C12 into the cells prior to starvation, and then starve in the presence of the inhibitors.

Re: As suggested, we have performed pulse-chase experiments. HepG2 cells were incubated with AP-C12 in DMEM containing 10% XerumFree (FBS replacement) for 1 h. After washing with HBSS+ three times, the cells were treated with 50 nM Baf-A1, 10 mM 3-MA, or 100 μ M DEUP in HEPES-buffered HBSS+ for 6 h in a CO₂ incubator and imaged. In DEUP-treated cells, distinct fluorescence signals remained in LDs in red color, while the others showed significantly diminished fluorescence both in the Green and Red channels. This result suggests that most of the AP-C12 metabolites, except for those incorporated into LDs in DEUP-treated cells, were extracellularly released during this period. The radioactive assay using [1-¹⁴C]palmitic acid, reported by Murphy et al. (*J. Neurosci. Res.*, 1992), showed rapid generation of the oxidized products within 10 min in mouse brain astrocytes. More recently, Uchinomiya et al. demonstrated that the β -oxidation product of a chemical probe bearing a fatty acid chain (FAO probe) was detected within 30 min in HepG2 under nutrition starvation conditions (*Chem. Commun.*, 2020, ref 13). Thus, the imaging results, *i.e.* the fluorescent signal of the AP dye in LDs was diminished even though the autophagy process was inhibited, can be explained by the fact that after lipolysis of TAG in LDs, AP-C12 was catabolized in mitochondria until the carbon chain was sufficiently shortened (e.g. AP-C6), which is then effluxed outside the cell. Such the rapid metabolic degradation of AP-C12 is one of the most important features compared to BODIPY 558/568 C12 that remains in mitochondria even after 24 h.

Although inhibition of the ATGL activity with DEUP effectively suppressed the degradation of LDs, we could not exclude the possibility that lipophagic degradation of LDs was involved during this process; that is, the small LDs may undergo the starvation-induced lipophagy before autophagy inhibitors took effect. As reported by Schott et al. (*J. Cell Biol.*, 2019, ref 47), lipolysis targets larger-sized LDs to reduce their size and is upstream of lipophagy. At this stage, although it is still unclear how lipolysis and

lipophagy act synergistically in lipid metabolism, we believe that AP-C12 will be a powerful tool to elucidate the degradation mechanism of LDs. We added these experimental results to the end of the section as follows. The detail procedures have been shown in the supporting information.

“On the other hand, in primary mouse hepatocytes, it has been reported that lipophagy-derived FAs from LDs are extracellularly effluxed outside the cell under serum deprivation conditions, and then the FAs are transported back into the cell and supplied into mitochondria⁵⁵. To avoid the autophagic degradation of LDs during pulsing, the cells were incubated with AP-C12 in DMEM containing 2% XerumFree (FBS replacement) for 1 h, and then chased in the absence or presence of Baf-A1, 3-MA or DEUP in HEPES-buffered HBSS (Figure S19). In the control and autophagy-inhibited cells, after the 6 h chase, significant decrease was observed, and negligible signals in fluorescence were detected both in the green and red channels, while in DEUP-treated cells, distinct fluorescence signals in LDs remained. AP-C12 is effectively oxidized in mitochondria and then released outside the cells, which seems to be shorter than the time required for induction of lipophagy; therefore, we cannot exclude the possibility of lipophagic degradation of LDs given the result of the ineffective inhibition by 3-MA. However, it is obvious that lipolysis is contributed to the degradation of relatively larger LDs under the HBSS-starved conditions. By accelerating autophagy beyond the rate that occurs under HBSS starvation using Rapa, many LDs increased in size and intensity were detected (Figures 4a and 4b, Rapa)⁵², revealing that autophagic process is also strongly related to the FA metabolism. Although it is still unclear how lipolysis and lipophagy are synergistically related to the degradation of LDs⁴⁵, the present results suggest that the inhibition of autophagy does indeed interfere with the supply of FAs to mitochondria, albeit that this is not the predominant mode of supply; rather, LD lipolysis by ATGL is an essential mechanism of their degradation.”

Supplementary Figure S19. Pulse-chase assay with AP-C12. HepG2 cells were pulsed with 5 μ M of AP-C12 in DMEM supplemented with XerumFree (TNC BIO) for 1 h. After the cells were rinsed with HBSS+ three times, chased in HEPES-buffered HBSS+ containing 50 nM Bafilomycin A1 (Baf-A1), 10 mM 3-methyladenine (3-MA), or 100 μ M diethylumbelliferyl phosphate (DEUP) for 6 h.

Minor Points

Line 16: FAs are not the sole precursors of most lipids. What about glycerol? Sphingosine?

Re: Thank you for the comment. We deleted this part.

Line 29: It is not clear what is meant by invasive. Why is imaging mass spectrometry or alkyne-labelled FAs more “invasive” than fluorescent probes? Do the authors mean those other techniques are not compatible with live cell-imaging?

Re: We sincerely apologize for our mistake. While imaging mass spectrometry is not capable of observing lipid distributions in living cells, Raman imaging allows non-invasive monitoring of the lipid dynamics. In general, however, fluorescence imaging has the advantage of high spatiotemporal resolution over these techniques. To emphasize this point, we have written this text to start with respect to spatiotemporal resolution.

Reviewer #3 (Remarks to the Author):

The manuscript by Kajiwara et al describes a fatty acid derivative (AP-C12) of a fluorescent dye presenting negative solvatochromism. It is shown that this probe can be involved in different metabolic pathways that can direct it to different organelles, where it changes the color because of differences in local polarity. A variety of inhibitors of enzymes were used to block some metabolic pathways of lipids, which allowed visualizing the response of the probe. Overall, the manuscript is well written, the idea to combine solvatochromism of a probe with its metabolic transformation is rather novel and the results obtained are interesting. The developed probe may find a number of applications to monitor distribution and metabolism of fatty acids and to screen relevant drugs. The technical quality of the work is good, although some additional quantification analysis should be provided. Moreover, the manuscript lacks appropriate positioning of the developed probe with respect to existing solvatochromic dyes and some additional experimental data should be provided to support the claims. Therefore, I recommend this manuscript for publication in Nature Communications after major revisions. The detailed comments are provided below.

Major issues:

1) There has been an intensive research on the design and application of solvatochromic dyes to visualize variations at the level of lipid membranes, organelles and lipid droplets. The examples include Nile Red and its derivatives (DOI: 10.1021/jacs.7b03846; 10.1021/jacs.0c10972; 10.1073/pnas.2016897118); push-pull pyrene (DOI: 10.1038/srep18870; 10.1021/acs.analchem.0c00023); and Laurdan (DOI: 10.1038/nprot.2011.419; 10.12688/f1000research.11577.2; doi.org/10.1073/pnas.1118288109), which present positive solvatochromism. All these probes revealed fine differences in the local polarity within the organelles (in line with those shown in this work) as well as showed sensitivity to the cellular processes. In particular, the effects of starvation, oxidative stress and autophagy have been already studied by the solvatochromic probes (e.g. DOI: 10.1021/acs.jpcclett.9b00668; 10.1039/D1AN00667C). All these works (and other relevant ones) should be carefully mentioned in the Introduction in order to properly position the present solvatochromic probe with respect to the current state of the art. Moreover, the authors should make comparison of the new probe with already reported solvatochromic probes in the

Conclusions section or end of Results and Discussion. Currently, this comparison is missing in the manuscript.

Re: Thank you for the suggestion. The distinct difference of AP-C12 from the present solvatochromic probe is that our probe is not designed to visualize the organelle polarity, but to monitor the FA metabolites distributed over the whole cell. We agree that the practical applications of the present solvatochromic probes to cell imaging, that is the changes in the membrane polarity have successfully been visualized during cellular events, should be mentioned in the Introduction. We added the following sentence on page 3.

“In particular, by using solvatochromic dyes, organelle membranes and LDs have been discriminated based on the differences in the local polarity arising from lipid compositions¹⁶. Moreover, solvatochromic fluorescent probes allow to monitor the changes in the lipid polarity induced by starvation¹⁷, oxidative stresses¹⁸, and autophagy¹⁹.”

16. Klymchenko, A. S. Solvatochromic and Fluorogenic Dyes as Environment-Sensitive Probes: Design and Biological Applications. *Acc. Chem. Res.* **50**, 366–375 (2017).

17. Ashoka, A. H., Ashokkumar, P., Kovtun, Y. P. & Klymchenko, A. S. Solvatochromic near-infrared probe for polarity mapping of biomembranes and lipid droplets in cells under stress. *J. Phys. Chem. Lett.* **10**, 2414–2421 (2019).

18. Danylchuk, D. I., Jouard, P.-H. & Klymchenko, A. S. Targeted solvatochromic fluorescent probes for imaging lipid order in organelles under oxidative and mechanical stress. *J. Am. Chem. Soc.* **143**, 912–924 (2021).

19. Liang, T., Qiang, T., Ren, L., Wanga, B. & Hu, W. An ultrasensitive polarity-specific two-photon probe for revealing autophagy in live cells during scrap leather-induced neuroinflammation process. *Analyst* **146**, 4659–4665 (2021).

The noticeable advantage of our negative solvatochromic dye over typical positive one is that the former emits strong fluorescence even in water. Because our goal is to image FA metabolism in living cells, the property that allows to detect the fluorescence signals in polar environment such as in cytosol and in mitochondrial matrix is of great importance. Regarding this point, we added the following sentence to the Result and Discussion on page 6.

“The strong fluorescence, even in water, is of great importance in visualizing the distribution of the dye throughout the cell, which is the most distinctive advantage over typical positive-solvatochromic dyes¹⁶.”

2) Page 7: “Incubation for a further 30 min resulted in an increase in the fluorescence signals from LDs in the red channel, while relatively weak fluorescence with no distinct features was observed in the green channel (Figure 2a, middle). This suggests that AP-C12 was taken up by the cells as an exogenous FA and readily converted into the corresponding TAGs in a similar manner to natural FAs, and that the TAGs subsequently accumulated in LDs.” This is actually the central claim of the present work, but there is no direct experimental support for it. The authors made an experiment with an inhibitor of DTAG to support indirectly this claim. However, to provide a solid support that the probe AP-C12 is really converted into TAG and thus accumulates inside LDs, the authors should make LC-MS spectral analysis from the cell extract with chloroform, similarly to that performed for analysis of the beta-oxidation products.

Re: We have tried to detect lipids extracted from cells by LS-MS. However, even for authentic samples (e.g. triolein standard), any peaks could not be detected with our instruments. Alternatively, we performed a chromatographic analysis of the AP-C12 metabolites to confirm that AP-C12 is incorporated into phospholipids or neutral lipids.

After treatment with AP-C12 for 6 h, trypsinized HepG2 cells were harvested and homogenized in CHCl₃/MeOH. The sample separated on a TLC plate was imaged by a fluorescence scanner (Typhoon FLA9500, GE HealthCare). In addition of original AP-C12, many spots corresponding to the metabolic products were newly observed. Based on the retention time of the authentic sample, AP-C6, as well as the reported results using radioactive FAs, the spots seen just below AP-C12 are ascribed to the β -oxidation products, whereas the spots appearing above AP-C12 could be attributed to TAGs and cholesterol ester containing at least one AP moiety. The lower spots with $R_f = 0.15$ and the origin may be from DAGs and phospholipids, respectively. This result is the strong evidence that AP-C12 is really converted into TAG and phospholipids as well as degraded by the β -oxidation. Regarding this point, we have added the TLC image to Figure 2d and the explanations to the main text as follows. The detail of the experimental procedure was shown in the Supporting Information.

“To confirm that AP-C12 is metabolically distributed in the cells, after HepG2 cells were incubated with the probe in HBSS+ for 6 h, the cell lysate was separated on a thin-layer chromatography (TLC) plate, and then imaged using a fluorescence scanner. In addition to the original spot of AP-C12 ($R_f = 0.48$), many spots corresponding to the metabolic products were newly observed (Figure 2d). Based on the retention time of the authentic sample, AP-C6, and the reported results using radioactive FAs³¹, the spots seen just below AP-C12 may correspond to the β -oxidation products, whereas the spots appearing above AP-C12 could be attributed to TAGs and cholesterol ester containing at least one AP moiety. Moreover, the lower plots with $R_f = 0.15$ and the origin may be from DAGs and phospholipids, respectively. This result strongly supports that the cell images observed with AP-C12 are from the metabolized probes localized in the respective organelle.”

3) Data analysis should be improved to provide more quantitative description of the solvatochromic response of the probes. At the moment the probes response is analyzed in form of ratiometric images and scatter plots, which do not really provide the values. The attempt to quantify the data is given in Figure 4d, but in this case, red and green channels are presented separately. However, the real advantage of solvatochromic dyes is a possibility to perform ratiometric data analysis, which are independent of local dye concentration. Therefore, the authors should provide systematically the values of green to red ratio and this should be done for the whole cell and some regions of interest (e.g. lipid droplets, ER and mitochondria, when they are well identified).

Re: Thank you for the suggestion. We agree that ratiometric analysis using solvatochromic probes is suitable for quantitatively evaluating the microenvironment regardless the dye concentration. In contrast to this, our analytical method using a 2D histogram can provide information about the signal intensities corresponding to the concentrations of the AP-C12 metabolites. In this case, each slope indicates the ratio of

green and red channels, which were determined from the ratio image shown in Figure 4c and Figures S14-S18. Indeed, lipid droplets and mitochondria show ~ 0.3 and ~ 1.2 of G/R ratios. The intermediate G/R ~ 0.7 may correspond to ER arising from DAG containing the AP moiety. To evaluate the effects of each reagent used in Figure 4, we analyzed the distribution of the G/R ratio in images and compared with that for the control. The data were added to Figures S14-S18. It is clear that the peak is shifted to the hydrophobic side when treated with DEUP.

Minor points:

- 1) Page 4: “negative solvatochromic properties” in the subtitle should be changed to “negative solvatochromism”.
- 2) Page 16: “polarity environment” should be “environment polarity”.
- 3) Page 19: Last paragraph is not clear because in the first sentence the authors mention “accelerating autophagy” and then, in next one, they talk about “inhibition of autophagy”. These two sequences should be better connected to improve clarity.
- 4) Conclusions section “fluorescent yields” should be “fluorescence quantum yields”. Moreover, it is not clear what is meant by “standard probes”. The latter issue is connected with the problem mentioned in point 1 of the major issues.

Re: Thank you very much. We have corrected all of the above as pointed out.

REVIEWER COMMENTS

Reviewer #2 (Remarks to the Author):

The revisions to the manuscript by Kajiwara and Osaki et al. successfully addressed a majority of the concerns identified in the original draft. Specifically, the analysis of AP-C12 incorporation into lipid species through their TLC analysis supports the ability of AP-C12 to be metabolized by the cell similar to “natural” FAs. The use of AP-C12 in pulse chase experiments further demonstrated the utility of this probe to assess the metabolism of incorporated lipids which greatly increases the potential uses of AP-12.

However, the authors were unable to address our concerns in relation to the localization and metabolic fate of AP-C12 with green spectral properties. Specifically, their main lines of evidence that AP-C12 which fluoresces green is localized to the mitochondrial matrix is based on 1) the colocalization of green fluorescence with a mitochondrial marker and 2) the known properties of AP-C12 in aqueous environment which mimic that of the mitochondrial matrix. This evidence alone is unable to distinguish whether AP-C12 which fluoresces green is accumulated within the matrix or on the cytoplasmic surface of mitochondria. To distinguish these possibilities, the authors inhibit FA transport into mitochondria using etomoxir an inhibitor of CPT-1. Treatment with etomoxir however did not cause a decrease in green fluorescence. The authors suggest in their revisions that this result may be due to confounding factors of etomoxir treatment, namely decreased TAG synthesis and increased phospholipid synthesis. This seems unlikely as their experiments in S12 indicate that etomoxir treatment did not alter LD size or number suggesting no change in TAG synthesis. Further, their spectral data does not match that of DGAT1 inhibition by T863 which shows a large increase in yellow fluorescence and decrease in red due to inhibition of TAG synthesis and increased phospholipid synthesis. Based on their data there are two more likely explanations: 1) etomoxir successfully inhibited transport of AP-C12 into the mitochondrial matrix and therefore the green fluorescence is not indicative of localization to the matrix, or 2) unlike Bodipy-C12, AP-C12 is not dependent on CPT-1 mediated transport (which has been observed for small chain FAs) and therefore is bypassing the trafficking methods used by endogenous FAs. If AP-C12 is bypassing CPT-1 mediated transport this may also be the cause of increased rates of β -oxidation in comparison to Bodipy-C12.

Due to the above concern, we suggest the text corrections indicated below. Given these changes, we would support publication in Nature Communications.

Major:

Page 1: The text indicating distribution into the polar mitochondrial matrix can be visualized should be removed.

Page 16: Text should be included to make clear that further work is needed to validate the localization of green fluorescent AP-C12 to the mitochondrial matrix and whether AP-C12 requires CPT-1 transport to traffic into the mitochondrial matrix. Alternatively this could be included in the conclusions as a limitation to the current study.

Minor:

Page 10: TLC experiments were performed after incubation in HBSS for 1hr rather than 6hr.

Page 10: The TLC experiments done by Bartz et al. in the included reference did not utilize radioactive FAs, rather endogenous levels of lipids were visualized. The text should be adjusted to reflect this.

Reviewer #3 (Remarks to the Author):

In the revised manuscript, the authors addressed well all my concerns and made additional new experiments and data analysis to support their claims. Now I can recommend this manuscript for publication in the present form.

Reviewer #2 (Remarks to the Author):

The revisions to the manuscript by Kajiwara and Osaki et al. successfully addressed a majority of the concerns identified in the original draft. Specifically, the analysis of AP-C12 incorporation into lipid species through their TLC analysis supports the ability of AP-C12 to be metabolized by the cell similar to “natural” FAs. The use of AP-C12 in pulse chase experiments further demonstrated the utility of this probe to assess the metabolism of incorporated lipids which greatly increases the potential uses of AP-12.

Thank you very much for evaluating our works.

However, the authors were unable to address our concerns in relation to the localization and metabolic fate of AP-C12 with green spectral properties. Specifically, their main lines of evidence that AP-C12 which fluoresces green is localized to the mitochondrial matrix is based on 1) the colocalization of green fluorescence with a mitochondrial marker and 2) the known properties of AP-C12 in aqueous environment which mimic that of the mitochondrial matrix. This evidence alone is unable to distinguish whether AP-C12 which fluoresces green is accumulated within the matrix or on the cytoplasmic surface of mitochondria. To distinguish these possibilities, the authors inhibit FA transport into mitochondria using etomoxir an inhibitor of CPT-1. Treatment with etomoxir however did not cause a decrease in green fluorescence. The authors suggest in their revisions that this result may be due to confounding factors of etomoxir treatment, namely decreased TAG synthesis and increased phospholipid synthesis. This seems unlikely as their experiments in S12 indicate that etomoxir treatment did not alter LD size or number suggesting no change in TAG synthesis. Further, their spectral data does not match that of DGAT1 inhibition by T863 which shows a large increase in yellow fluorescence and decrease in red due to inhibition of TAG synthesis and increased phospholipid synthesis. Based on their data there are two more likely explanations: 1) etomoxir successfully inhibited transport of AP-C12 into the mitochondrial matrix and therefore the green fluorescence is not indicative of localization to the matrix, or 2) unlike Bodipy-C12, AP-C12 is not dependent on CPT-1 mediated transport (which has been observed for small chain FAs) and therefore is bypassing the trafficking methods used by endogenous FAs. If AP-C12 is bypassing CPT-1 mediated transport this may also be the cause of increased rates of β -oxidation in comparison to Bodipy-C12.

Due to the above concern, we suggest the text corrections indicated below. Given these changes, we would support publication in Nature Communications.

Re: Thank you very much for providing detailed suggestions. We believe that the observation of the β -oxidation products of AP-C12 is strong evidence that AP-C12 is localized in the mitochondrial matrix. As the reviewer pointed out, due to the low inhibition effect of etomoxir, it is also possible that AP-C12 could bypass the CPT-1-mediated transport and directly enter the mitochondrial matrix. In this case, AP-C12 may act as a medium-chain fatty acid with a carbon length of about 12, which is known to bypass the carnitine cycle. However, we cannot completely exclude the possibility of CPT-1-mediated transportation, and it may take place in both pathways. Thus, we wrote the following sentence in the conclusions:

“However, at current study, the mitochondrial localization mechanism of AP-C12, *i.e.*, whether CPT-1 is required for the transportation of AP-C12 into the mitochondrial matrix, is not clear, and therefore further work is needed to validate this point.”

Major:

Page 1: The text indicating distribution into the polar mitochondrial matrix can be visualized should be removed.

Re: Whether via CPT-1 or direct uptake, AP-C12 and its metabolites should be present into the mitochondrial matrix. However, since the polar region is not limited to the mitochondrial matrix, this sentence has been revised as follows.

“Owing to the negative-solvatochromism of this AP dye, the distribution of the metabolically incorporated FA probe in non-polar lipid droplets (LDs), medium-polarity membranes, and the polar aqueous regions including cytosol and mitochondrial matrix, can be visualized in different colors.”

Page 16: Text should be included to make clear that further work is needed to validate the localization of green fluorescent AP-C12 to the mitochondrial matrix and whether AP-C12 requires CPT-1 transport to traffic into the mitochondrial matrix. Alternatively this could be included in the conclusions as a limitation to the current study.

Re: We added the necessity of the further work to the conclusions, as mentioned above.

Minor:

Page 10: TLC experiments were performed after incubation in HBSS for 1hr rather than 6hr.

Re: We thought that the TLC experiment after 6 h would be sufficient to prove the metabolic incorporations of AP-C12. In addition to this, we also conducted an additional “1 h” experiment. Although we could detect the metabolic products as shown below, the intensities of each peak are much weaker than those observed after 6 h incubation. As we mentioned in the previous response sheet, since AP-C6 and AP-C8 are also soluble in water, the TLC results are not comparable with the LC-MS analysis. This result seems not to be of crucial importance for this manuscript to be included in the main text and Supplementary Information, so it is shown only for the reviewers.

Page 10: The TLC experiments done by Bartz et al. in the included reference did not utilize radioactive FAs, rather endogenous levels of lipids were visualized. The text should be adjusted to reflect this.

Re: Thank you for the suggestion. We corrected it as noted.

“, and the reported lipid chromatography of LDs isolated from cells³¹,”

Reviewer #3 (Remarks to the Author):

In the revised manuscript, the authors addressed well all my concerns and made additional new experiments and data analysis to support their claims. Now I can recommend this manuscript for publication in the present form.

Thank you very much for evaluating our works.